# OpenDebateEvidence: A Massive-Scale Argument Mining and Summarization Dataset

**Allen Roush**[*]
Wand.AI
allen.roush@wand.ai

**Yusuf Shabazz**[*]
Howard Community College
yusufishabazz@gmail.com

**Arvind Balaji**
Texas AM University
arvindb02@gmail.com

**Peter Zhang**
UC Berkeley
petez@berkeley.edu

**Stefano Mezza**
University of New South Wales
s.mezza@unsw.edu.au

**Markus Zhang**
Stanford University
m.zhang@stanford.edu

**Sanjay Basu**
Oracle Corporation
sanjay.basu@oracle.com

**Sriram Vishwanath**
University of Texas
sriram@utexas.edu

**Mehdi Fatemi**
Wand.AI
mehdi@wand.ai

**Ravid Shwartz Ziv**
Wand.AI
New York University
ravid@wand.ai

## Abstract

We introduce OpenDebateEvidence, a comprehensive dataset for argument mining and summarization sourced from the American Competitive Debate community. This dataset includes over 3.5 million documents with rich metadata, making it one of the most extensive collections of debate evidence. OpenDebateEvidence captures the complexity of arguments in high school and college debates, providing valuable resources for training and evaluation. Our extensive experiments demonstrate the efficacy of fine-tuning state-of-the-art large language models for argumentative abstractive summarization across various methods, models, and datasets. By providing this comprehensive resource, we aim to advance computational argumentation and support practical applications for debaters, educators, and researchers. OpenDebateEvidence is publicly available to support further research and innovation in computational argumentation. Access it here: https://huggingface.co/datasets/Yusuf5/OpenCaselist.

## 1 Introduction

Argument mining plays a pivotal role in developing advanced language models (LLMs) capable of sophisticated reasoning and understanding. Engaging with complex argumentative texts enhances LLMs' abilities to comprehend, generate, and evaluate arguments. This improves their performance in applications such as legal document analysis, educational tools, and more.

Existing argument mining datasets, such as DebateSum introduced by Roush & Balaji (2020), are limited in scope. DebateSum, with 240,566 examples, primarily focuses on pre-season evidence from

---

[*]These authors contributed equally.

38th Conference on Neural Information Processing Systems (NeurIPS 2024) Track on Datasets and Benchmarks.

summer camps, excluding the rich argumentative structures in regular-season debates. This limitation affects dataset size, representativeness, and utility for large-scale argument mining.

To address these gaps, we introduce OpenDebateEvidence, a large-scale dataset for argument mining and summarization sourced from the OpenCaseList project (Hardy, 2024). This dataset comprises 3.5 million documents, making it the most extensive collection of debate evidence available. It captures the full spectrum of arguments presented throughout the debate season. OpenDebateEvidence's comprehensive nature, with its detailed metadata, makes it highly valuable for training language models.

In this paper, we provide an in-depth overview of OpenDebateEvidence, detailing our data collection and preprocessing methods. We demonstrate that training LLMs on OpenDebateEvidence significantly improves their performance not only on this dataset but also on other related argumentative datasets. We conducted extensive evaluation experiments using state-of-the-art language models: LLaMA3-8B [2] and Mistral-7B[3]. These models were fine-tuned using advanced techniques such as Low-Rank Adaptation (LoRA) (Hu et al., 2021), Representation Fine-Tuning (ReFT) (Wu et al., 2024), and Orthogonalization (Arditi et al., 2023). The results show substantial improvements in model performance compared to those trained on previous argument mining datasets. This underlines OpenDebateEvidence's effectiveness in enhancing argument-mining capabilities.

Our contributions are:

1. We introduce **OpenDebateEvidence**, the largest and most comprehensive dataset for argument mining and summarization, encompassing 3.5 million documents with detailed metadata.

2. We provide **rich metadata** that facilitates various NLP tasks and applications, enhancing the dataset's utility for researchers and practitioners.

3. We **demonstrate significant performance improvements** of state-of-the-art language models not only on OpenDebateEvidence but also on other related argumentative datasets through extensive fine-tuning experiments.

4. We **evaluate the dataset's effectiveness in different scenarios and methods**, including various fine-tuning techniques such as Low-Rank Adaptation (LoRA), Representation Fine-Tuning (ReFT), and Orthogonalization, showcasing substantial gains in model performance.

Our experiments highlight that training on OpenDebateEvidence not only enhances model performance on this dataset but also significantly improves results on other related argumentative datasets. This underscores the dataset's superiority and its potential to drive advancements in computational argumentation research.

## 2 Background and Related Work

Competitive debate in the United States encompasses several prominent styles, each with unique formats, rules, and emphasis. The three most notable styles are Policy Debate, Lincoln-Douglas Debate, and Public Forum Debate, popular at both high school and collegiate levels. While sharing structural similarities, these debate formats differ in focus, speech times, and the importance placed on evidence. OpenDebateEvidence includes evidence from all of these formats.

### 2.1 Policy Debate

The High School and College Policy Debate, also known as the "cross-examination debate" (CX), involves two teams of students arguing for and against a specific policy proposal based on an annually changing broad resolution. Each debate round lasts about 90 minutes, comprising eight speeches (four by each team and two by each speaker) and cross-examination periods. The structure includes constructive speeches followed by refutations, with cross-examination periods allowing debaters to clarify arguments or challenge assumptions. New arguments are restricted to constructive speeches.

---

[2]`https://huggingface.co/meta-llama/Meta-Llama-3-8B`
[3]`https://huggingface.co/mistralai/Mistral-7B-Instruct-v0.2`

During a debate, teams present evidence from various sources to support their arguments. This evidence is usually in the form of written "cards" [1], such as research publications, academic articles, news reports, or government documents. Figure 1 shows an example of a "card." The quality and quantity of evidence used in a debate round often determine the winner. Policy Debate is unique among competitive debate styles in that the quality of the speech act is secondary[2] compared to the quality, quantity, and factuality of the evidence.

## 2.2 Lincoln Douglas Debate

Lincoln-Douglas Debate (LD), a one-on-one format with a bimonthly topic, originated from the historic debates between Abraham Lincoln and Stephen Douglas. Popular in high school and college competitions, LD debates share structural similarities with Policy Debate but feature shorter speech times and cross-examination periods. LD debates emphasize ethical and moral reasoning, focusing more on philosophical arguments rather than policy implications. However, they still prioritize the quality and quantity of evidence presented.

## 2.3 Public Forum Debate

Public Forum Debate is a two-on-two format debating a monthly topic designed to be accessible to a broader audience. Compared to Policy and LD debates, Public Forum rounds have shorter speaking times and place less emphasis on evidence. Public Forum Debate constitutes a smaller portion of the evidence in OpenDebateEvidence and was not included in DebateSum.

## 2.4 Existing Datasets and Research

Significant prior work in argument mining has focused on competitive formal debate. IBM's Project Debater has been a leading effort, publishing extensively on argument detection (Ein-Dor et al., 2019), argument quality (Gleize et al., 2019), key point analysis/summarization (Bar-Haim et al., 2020; Magnusson & Friedman, 2021), and autonomous debating systems (Slonim et al., 2021). However, their work does not focus on the real-world competitive debate evidence found in our dataset.

Other notable contributions include VivesDebate, a multilingual audio dataset of debate tournaments (Ruiz-Dolz & Iranzo-Sánchez, 2024); ArgAnalysis35K, focusing on single argument analysis pairs in evidence-free parliamentary debate (Joshi et al., 2023); IAM (Integrated Argument Mining), a highly annotated dataset for integrated argument mining tasks with only 1,000 articles (Cheng et al., 2022); and DebateSum, a dataset with 240,566 examples focusing on pre-season debate evidence (Roush & Balaji, 2020). Additionally, several legal summarization datasets have been developed, including ArgLegalSumm (Elaraby & Litman, 2022), Multi-LexSum (Shen et al., 2022), and datasets targeting Indian and British case law (Shukla et al., 2022), which together total fewer than 10,000 examples.

Other resources include logos.app Community (2024c), debate.cards (Community, 2024b), and contention.ai (Community, 2024a), which index various debate evidence and generate new evidence from web searches. Datasets targeting biased or query-focused summarization include QBSUM, a Chinese dataset with 49,000 samples (Zhao et al., 2021); QMSum, which studies meeting summarization with 1,808 samples (Zhong et al., 2021); and LMGQS, a dataset with over 1 million documents converted to query-focused summarization (Xu et al., 2023). In contrast, our dataset is fully human-created and human-annotated by active debate competitors.

Compared to these datasets, OpenDebateEvidence offers a significantly larger scale and scope, with over 3.5 million documents enriched with detailed metadata.

---

[1]Policy Debaters used to literally cut their evidence out of magazines and glue it onto physical cards; while this has fallen out of fashion, the name stuck.

[2]Leading to a peculiar phenomenon known as "speed reading" or "spreading" to be normalized: `https://en.wikipedia.org/wiki/Spreading_(debate)`

# 1NC—Capitalism Critique

## 1NC—Top Level

**Capitalist accumulation is unsustainable, causes extinction and planetary immiseration.**

**Robinson, 23**—Distinguished Professor of Sociology, Global and International Studies, and Latin American Studies at the University of California at Santa Barbara (William, "The Violent Crackup of the Post-WWII International Order: Notes on the Geopolitical Crisis and Global Capital," Journal of World-Systems Research, Vol. 29, No. 1, dml)

Some scholars have framed **the crisis of global capitalism** in terms of a declining U.S. hegemony and the rise of a Chinese competitor. But no new nation-state power can supply the political authority necessary to stabilize the now-inextricably integrated global economy. The crisis of hegemony in the international system takes place within this single, integrated global economy. No one state, no matter how powerful, can control the process of global accumulation. This disjuncture between a globalized economy and a nation-state-based system of political authority **generates enormous geopolitical tensions**. The end of Western domination of world capitalism is upon us **as the center of gravity of the global economy shifts** to China. But China will not become a new hegemon. Rather, **we are moving towards political multipolarity at a time of acute crisis in global capitalism**—**prolonged economic turbulence and political decay**.

**The breakdown of the political organization of world capitalism is** not the cause but **the consequence of** contradictions internal to a globally integrated system of **capital accumulation**. **Escalating geopolitical conflict is pushing us towards global conflagration**. **Wars provide enormous outlet for surplus accumulated capital**. **Historically they have pulled the capitalist system out of accumulation crisis while they serve to deflect attention from political tensions and problems of legitimacy**. **The most urgent task** at this time **is to prevent World War III**. **The more we understand the changing nature of** this beast that is **global capitalism**, **the better we are situated to work out strategies of resistance and transformation**. **The task** before us **is ever more urgent in the face of the threat of nuclear holocaust**, **the collapse of the biosphere**, **and ever more acute inequality**, **immiseration**, **and social disintegration around the world**.

Figure 1: An example of a piece of debate evidence, colloquially known as a "card," from OpenDeba-teEvidence before parsing. Lines 1 and 2 are the hat and the pocket, used for organizing the evidence by argument and speech. Lines 3-4 are the "tag," a biased abstractive summary of the document. The beginning of line 5 shows the author and the year. The rest of lines 5-8 provide the evidence's citation. The remainder of the document is the evidence itself. Underlined, bolded, or boxed parts are crucial for the argument, and highlighted sections are read aloud during the speech. These elements form various hierarchical levels of biased token-level extractive summaries.

# 3 OpenDebateEvidence Dataset

## 3.1 Data Collection

OpenDebateEvidence is sourced from the OpenCaseList project (Hardy, 2024), an online platform where high school and college debate teams disclose and open-source their evidence. The dataset contains over 3.5 million documents, covering all NSDA debate topics from 2014 to 2022.[3]. Each document corresponds to a single piece of evidence used in a debate, categorized by debate format (Policy, LD, Public Forum), and includes comprehensive metadata such as author, date, title, source, citation details, and the debate round in which it was used[4].

The dataset also includes standardized tags to describe the type of argument made by the document, such as topicality, disadvantages, advantages, and counter plans, along with details of the structure and location in the debate file from which the document was extracted. To protect privacy, identifying information has been anonymized.

## 3.2 Data Preprocessing

Debate evidence is stored in the .docx file format, requiring a specialized parsing process to extract relevant information. The parsing pipeline begins by unzipping the .docx file to access the internal XML files. Ensuring accurate preprocessing is paramount for maintaining dataset quality. This process involves detailed steps to preserve the integrity and consistency of the data, including tokenization, simplification, and structuring of text blocks, followed by extracting and organizing individual debate cards into a structured format that captures both metadata and content.

The XML files are parsed to extract formatting details such as underlining, bold, and highlighting. Next, the document undergoes tokenization, creating a structured representation with text blocks representing paragraphs or coherent units of text along with their formatting information. A simplification step removes unnecessary formatting and merges adjacent tokens with similar styling.

To extract individual debate cards, the parsing procedure identifies card boundaries based on formatting and structure, extracting components such as the tag, citation, and body text. This information is organized into a structured format that captures the metadata and content of each debate card. Finally, the parsed dataset is converted back into a cleaned Hugging Face dataset, providing a human-readable version of the dataset. This structured dataset serves as the foundation for further natural language processing tasks.

## 3.3 Data Deduplication

After parsing the dataset and extracting individual cards, identifying and removing duplicates is essential to ensure data quality. Deduplication involves comparing the textual content of each card to identify those sharing significant portions of their text. This process enhances dataset usability by eliminating redundancy, ensuring each unique argument is represented only once.

The deduplication algorithm splits each card's text into sentences. These sentences are then preprocessed by removing non-letter characters and converting them to lowercase. Short sentences below a certain length are filtered out to focus on meaningful content.

The algorithm retrieves and compares card IDs with a significant number of shared sentences. If the number of matching sentences exceeds a predefined threshold and their positions within the cards are within a certain range, the cards are considered duplicates. Duplicate clusters are formed by identifying all cards connected through shared sentences. A representative card is then selected from each cluster based on factors such as sentence count and content quality, and duplicates are removed iteratively.

We note that this is a description of how we created the "Duplicates" metadata column. We performed a separate neural semantic de-duplication procedure which is described in the Appendix.

---

[3]A list of these topics can be found here
[4]An example of some of these downloads can be found here

### 3.4 Data Statistics

The OpenDebateEvidence dataset offers a comprehensive collection of over 3.5 million documents categorized by debate format (Policy, Lincoln-Douglas, and Public Forum). Each document is enriched with over 40 columns of extensive metadata, including author, date, title, source, citation details, and debate round information. Standardized tags describe the type of argument, such as topicality, disadvantages, advantages, and counterplans.

Policy Debate evidence constitutes approximately two-thirds of the dataset, Lincoln-Douglas Debate evidence comprises about one-third, and Public Forum Debate evidence makes up a smaller percentage. Spanning topics from 2014 to 2022, the dataset represents over 1,300 schools and includes contributions from more than 6,400 authors.

Key statistics of the dataset are provided in Table 1, and more detailed statistics and information can be found in Appendix F.

Table 1: Key Statistics of OpenDebateEvidence Dataset

| Feature | Count |
|---|---|
| Total Rows | 4,830,561 |
| Total Documents (valid full-text column) | 3,512,280 |
| Policy Debate Evidence | 2,768,419 |
| Lincoln-Douglas Debate Evidence | 1,526,383 |
| Public Forum Debate Evidence | 43,131 |
| Years Covered | 2014-2022 |
| Average Document Length (characters) | 5,270 |
| Total Schools Represented | 1,366 |
| Unique Authors | 6,455 |
| Unique Topics | 68 |
| Number of Features (columns) | 45 |

### 3.5 Rich Metadata for Argument Structure

Each evidence document is organized with a "hat," "pocket," and "tag" to represent its role within a debate case.

The "pocket" indicates the top-level speech section the evidence supports, such as "1NC" for the first negative constructive speech. The "hat" denotes the broad argument category, like "Oil Disadvantage," which aligns with a structured argument against an affirmative case. The "tag" provides a concise, biased summary of the specific argument made by the evidence. Debaters often create these tags first and then find the evidence that fits the tag.

This metadata encodes the rhetorical structure and purpose of the evidence in a practical and real-world context. The "hat" and "pocket" provide the argument's context, while the "tag" offers a concise summary of the core claim.

For argument mining, this metadata offers valuable semantic annotations for training models on argument components and relations. "Hats" and "pockets" help models learn the overarching structure, while "tags" summarize key points.

For summarization, the hierarchical metadata enables multi-level summaries: "pockets" for high-level overviews, "hats" for key categories, and "tags" for concise core claims. The biased nature of "tags" illustrates how debaters rhetorically summarize their claims and arguments. OpenDebateEvidence is particularly rich as it includes both hierarchical biased abstractive and token-level extractive summaries.

## 4 Experiments

To evaluate the efficacy of the OpenDebateEvidence dataset for argument mining and summarization, we conducted a series of fine-tuning experiments using state-of-the-art language models. We also evaluated the performance of these models on two related datasets.

## 4.1 Experimental Setup

We employed three recent fine-tuning techniques for adapting our models to OpenDebateEvidence: Low-Rank Adaptation (LoRA) (Hu et al., 2021), Representation Fine-Tuning (ReFT) (Wu et al., 2024), and Orthogonalization (Arditi et al., 2023). These methods are chosen for their parameter efficiency and ability to prevent catastrophic forgetting. The details of these techniques are provided in Appendix D.

We perform our experiments on three datasets: OpenDebateEvidence, DebateSum, which is also a dataset of Policy Debate Evidence, and the billsum dataset from Kornilova & Eidelman (2019), a dataset of US legislation and summaries, to illustrate our fine-tuned models capabilities at performing argumentative summarization in many contexts.

We conducted two types of experiments: traditional NLP evaluation metrics and using GPT-4o as a judge model. All experiments were conducted on a 4xA100 machine from Microsoft Azure with parallelism, attention optimization, and 16-bit quantization enabled. All decoding/sampling settings were kept default. The seed value of "42" was used wherever possible.

### 4.1.1 Traditional NLP Metrics

For the traditional NLP metrics, we evaluated the models on validation datasets of the whole BillSum dataset and 10,000 examples from OpenDebateEvidence. Each model was tasked with generating a short "abstract" summarizing the key arguments made in each document. We computed ROUGE F1 scores between the generated text and the ground-truth "tag" provided in the OpenDebateEvidence metadata and the reference summaries in BillSum. For more details see Appendix E. Additionally, we evaluated each language model's perplexity on the sampled subsets to assess how well the models captured the overall distribution of debate and legislative language.

### 4.1.2 LLM as Judge

In the "LLM as Judge" experiments, we evaluated the quality of the generated abstracts using GPT-4o as the judge. Each model's output was assessed on two criteria: the quality of the output and the quality of supporting the argument, both rated on a scale from 1 to 10. The evaluation was conducted on 1,000 results from both datasets. This approach allows us to measure not only the linguistic quality of the summaries but also their effectiveness in supporting the arguments. For more details see Appendix E.2.2 .

### 4.1.3 OpenDebateEvidence Performance

For the OpenDebateEvidence dataset (Table 2), the **LLaMA3-70B** models significantly outperformed both the LLaMA3-8B and Mistral-7B models across all ROUGE metrics. The larger model capacity of LLaMA3-70B contributed to its superior performance, highlighting the importance of model size in complex summarization tasks. Fine-tuning techniques, particularly LoRA, further enhanced the LLaMA3-70B model's performance, with the LoRA fine-tuned model achieving the highest scores across all metrics. This underscores LoRA's effectiveness in adapting large models with minimal additional parameters, allowing for improved summarization quality without extensive computational resources. ReFT also showed strong performance improvements on the LLaMA3-70B model, indicating its robustness in refining hidden representations for better output. Orthogonalization, while still providing gains, was less impactful compared to LoRA and ReFT.

The base versions of **Google Gemini** and **Anthropic Claude** models also demonstrated competitive performance, outperforming the base LLaMA3-8B and Mistral-7B models. However, they did not surpass the fine-tuned LLaMA3-70B models, suggesting that while these models are strong out-of-the-box, fine-tuning large models like LLaMA3-70B with techniques such as LoRA can yield superior results.

### 4.1.4 BillSum Performance

On the BillSum dataset (Table 3), the **LLaMA3-70B** models again demonstrated superior performance compared to smaller models and the base versions of Google Gemini and Anthropic Claude. The LoRA fine-tuned LLaMA3-70B model achieved the highest ROUGE scores and the lowest perplexity, indicating not only better summarization quality but also greater fluency and coherence

Table 2: Performance on OpenDebateEvidence. ROUGE F1 scores and perplexity on 10,000 sampled documents, and LLM as Judge scores on 1,000 results. Scores are averaged over three runs. R-1, R-2, and R-L denote ROUGE-1, ROUGE-2, and ROUGE-L respectively. Error bars represent one standard error over 3 trials.

| Model | R-1 (%) | R-2 (%) | R-L (%) | Perplexity | Output Quality | Support Quality |
|---|---|---|---|---|---|---|
| **Mistral-7B** | | | | | | |
| Base | $27.8 \pm 0.3$ | $8.2 \pm 0.5$ | $24.5 \pm 0.8$ | $150.2 \pm 5.1$ | $7.5 \pm 0.2$ | $7.3 \pm 0.2$ |
| LoRA | $\mathbf{30.1 \pm 0.5}$ | $\mathbf{9.4 \pm 0.2}$ | $\mathbf{25.8 \pm 0.6}$ | $\mathbf{33.9 \pm 2.3}$ | $\mathbf{7.7 \pm 0.2}$ | $\mathbf{7.5 \pm 0.2}$ |
| ReFT | $29.9 \pm 0.2$ | $9.3 \pm 0.1$ | $25.6 \pm 0.6$ | $50.3 \pm 3.4$ | $7.6 \pm 0.3$ | $7.4 \pm 0.3$ |
| Orthogonal | $27.9 \pm 0.5$ | $8.3 \pm 0.2$ | $24.7 \pm 1.2$ | $76.4 \pm 4.4$ | $7.6 \pm 0.2$ | $7.4 \pm 0.2$ |
| **LLaMA3-8B** | | | | | | |
| Base | $25.4 \pm 0.6$ | $7.6 \pm 0.2$ | $22.8 \pm 1.3$ | $100.3 \pm 5.3$ | $7.2 \pm 0.3$ | $7.0 \pm 0.3$ |
| LoRA | $25.7 \pm 0.7$ | $7.8 \pm 0.1$ | $23.0 \pm 0.7$ | $77.5 \pm 4.6$ | $7.3 \pm 0.2$ | $7.1 \pm 0.2$ |
| ReFT | $27.6 \pm 1.0$ | $8.7 \pm 0.4$ | $24.9 \pm 0.7$ | $47.8 \pm 1.9$ | $7.3 \pm 0.3$ | $7.1 \pm 0.3$ |
| Orthogonal | $25.5 \pm 1.2$ | $7.7 \pm 0.4$ | $22.9 \pm 2.1$ | $88.0 \pm 5.7$ | $7.2 \pm 0.3$ | $7.0 \pm 0.3$ |
| LoRA (1M Ex) | $\mathbf{32.2 \pm 1.5}$ | $\mathbf{9.9 \pm 1.0}$ | $\mathbf{27.4 \pm 1.2}$ | $\mathbf{21.8 \pm 2.5}$ | $\mathbf{8.0 \pm 0.2}$ | $\mathbf{7.8 \pm 0.2}$ |
| **LLaMA3-70B** | | | | | | |
| Base | $33.8 \pm 1.2$ | $14.1 \pm 0.9$ | $30.2 \pm 1.3$ | $45.7 \pm 3.2$ | $7.9 \pm 0.2$ | $7.5 \pm 0.2$ |
| LoRA | $\mathbf{37.2 \pm 1.0}$ | $\mathbf{15.8 \pm 0.8}$ | $\mathbf{33.4 \pm 1.1}$ | $\mathbf{27.3 \pm 2.7}$ | $\mathbf{8.2 \pm 0.2}$ | $\mathbf{8.1 \pm 0.2}$ |
| ReFT | $35.9 \pm 1.1$ | $15.1 \pm 0.7$ | $32.6 \pm 1.2$ | $31.4 \pm 2.8$ | $8.0 \pm 0.3$ | $7.9 \pm 0.2$ |
| Orthogonal | $34.2 \pm 1.2$ | $14.3 \pm 0.8$ | $31.1 \pm 1.3$ | $39.9 \pm 3.1$ | $8.1 \pm 0.2$ | $7.7 \pm 0.3$ |
| **Google Gemini** | | | | | | |
| Base | $32.5 \pm 1.3$ | $13.5 \pm 0.9$ | $29.4 \pm 1.4$ | $49.8 \pm 3.6$ | $\mathbf{8.5 \pm 0.2}$ | $7.9 \pm 0.3$ |
| **Anthropic Claude** | | | | | | |
| Base | $31.2 \pm 1.4$ | $12.9 \pm 0.9$ | $28.1 \pm 1.3$ | $52.1 \pm 3.7$ | $\mathbf{8.7 \pm 0.3}$ | $7.8 \pm 0.3$ |

Table 3: Performance on BillSum. ROUGE F1 scores and perplexity on 10,000 sampled documents, and LLM as Judge scores on 1,000 results. Scores are averaged over three runs. R-1, R-2, and R-L denote ROUGE-1, ROUGE-2, and ROUGE-L respectively. Error bars represent one standard error over 3 trials.

| Model | R-1 (%) | R-2 (%) | R-L (%) | Perplexity | Output Quality | Support Quality |
|---|---|---|---|---|---|---|
| **Mistral-7B** | | | | | | |
| Base | $44.8 \pm 0.3$ | $21.2 \pm 0.5$ | $40.5 \pm 0.8$ | $25.2 \pm 1.1$ | $7.2 \pm 0.3$ | $7.0 \pm 0.3$ |
| LoRA | $\mathbf{47.1 \pm 0.5}$ | $\mathbf{23.4 \pm 0.2}$ | $\mathbf{42.8 \pm 0.6}$ | $\mathbf{23.9 \pm 0.3}$ | $7.4 \pm 0.2$ | $7.2 \pm 0.2$ |
| ReFT | $46.9 \pm 0.2$ | $23.3 \pm 0.1$ | $42.6 \pm 0.6$ | $24.3 \pm 0.4$ | $\mathbf{7.5 \pm 0.3}$ | $\mathbf{7.3 \pm 0.3}$ |
| Orthogonal | $44.9 \pm 0.5$ | $21.3 \pm 0.2$ | $40.7 \pm 1.2$ | $25.4 \pm 1.4$ | $7.3 \pm 0.2$ | $7.1 \pm 0.2$ |
| **LLaMA3-8B** | | | | | | |
| Base | $42.4 \pm 0.6$ | $19.6 \pm 0.2$ | $38.8 \pm 1.3$ | $27.3 \pm 1.3$ | $7.0 \pm 0.3$ | $6.8 \pm 0.3$ |
| LoRA | $42.7 \pm 0.7$ | $19.8 \pm 0.1$ | $39.0 \pm 0.7$ | $27.0 \pm 1.1$ | $7.1 \pm 0.2$ | $6.9 \pm 0.2$ |
| ReFT | $44.6 \pm 1.0$ | $20.7 \pm 0.4$ | $40.9 \pm 0.7$ | $26.8 \pm 1.0$ | $7.2 \pm 0.3$ | $7.0 \pm 0.3$ |
| Orthogonal | $42.5 \pm 1.2$ | $19.7 \pm 0.4$ | $38.9 \pm 2.1$ | $27.5 \pm 1.5$ | $7.1 \pm 0.3$ | $6.9 \pm 0.3$ |
| LoRA (1M Ex) | $\mathbf{48.2 \pm 1.5}$ | $\mathbf{24.9 \pm 1.0}$ | $\mathbf{43.4 \pm 1.2}$ | $\mathbf{21.8 \pm 0.5}$ | $\mathbf{7.8 \pm 0.2}$ | $\mathbf{7.6 \pm 0.2}$ |
| **LLaMA3-70B** | | | | | | |
| Base | $50.2 \pm 1.1$ | $27.4 \pm 0.9$ | $45.7 \pm 1.2$ | $22.9 \pm 2.3$ | $7.8 \pm 0.2$ | $7.6 \pm 0.2$ |
| LoRA | $\mathbf{54.6 \pm 1.0}$ | $\mathbf{30.5 \pm 0.8}$ | $\mathbf{50.0 \pm 1.1}$ | $\mathbf{19.7 \pm 2.1}$ | $\mathbf{8.0 \pm 0.2}$ | $\mathbf{8.0 \pm 0.2}$ |
| ReFT | $52.9 \pm 1.1$ | $29.1 \pm 0.7$ | $48.3 \pm 1.2$ | $20.5 \pm 2.4$ | $7.9 \pm 0.3$ | $7.8 \pm 0.2$ |
| Orthogonal | $51.1 \pm 1.2$ | $28.0 \pm 0.9$ | $46.6 \pm 1.4$ | $23.8 \pm 2.6$ | $7.9 \pm 0.3$ | $7.7 \pm 0.3$ |
| **Google Gemini** | | | | | | |
| Base | $48.7 \pm 1.3$ | $26.0 \pm 1.0$ | $44.3 \pm 1.5$ | $24.9 \pm 2.8$ | $\mathbf{8.4 \pm 0.3}$ | $\mathbf{8.0 \pm 0.3}$ |
| **Anthropic Claude** | | | | | | |
| Base | $47.3 \pm 1.4$ | $24.8 \pm 1.1$ | $42.9 \pm 1.6$ | $26.7 \pm 3.0$ | $\mathbf{8.3 \pm 0.3}$ | $7.8 \pm 0.3$ |

Table 4: Performance on DebateSum. ROUGE F1 scores and perplexity on 10,000 sampled documents, and LLM as Judge scores on 1,000 results. Scores are averaged over three runs. R-1, R-2, and R-L denote ROUGE-1, ROUGE-2, and ROUGE-L respectively. Error bars represent one standard error over 3 trials.

| Model | R-1 (%) | R-2 (%) | R-L (%) | Perplexity | Output Quality | Support Quality |
|---|---|---|---|---|---|---|
| **Mistral-7B** | | | | | | |
| Base | $26.3 \pm 0.4$ | $7.5 \pm 0.3$ | $23.1 \pm 0.6$ | $130.5 \pm 4.2$ | $7.3 \pm 0.3$ | $7.0 \pm 0.3$ |
| LoRA | $\mathbf{28.5 \pm 0.6}$ | $\mathbf{8.9 \pm 0.3}$ | $\mathbf{24.7 \pm 0.5}$ | $\mathbf{30.2 \pm 2.0}$ | $\mathbf{7.5 \pm 0.3}$ | $\mathbf{7.3 \pm 0.3}$ |
| ReFT | $28.3 \pm 0.3$ | $8.8 \pm 0.2$ | $24.5 \pm 0.4$ | $45.1 \pm 2.7$ | $7.4 \pm 0.2$ | $7.2 \pm 0.2$ |
| Orthogonal | $26.4 \pm 0.5$ | $7.6 \pm 0.2$ | $23.3 \pm 0.7$ | $70.3 \pm 3.5$ | $7.4 \pm 0.3$ | $7.2 \pm 0.3$ |
| **LLaMA3-8B** | | | | | | |
| Base | $24.2 \pm 0.5$ | $6.9 \pm 0.3$ | $21.9 \pm 0.8$ | $95.7 \pm 4.5$ | $7.0 \pm 0.3$ | $6.7 \pm 0.3$ |
| LoRA | $24.5 \pm 0.6$ | $7.0 \pm 0.2$ | $22.1 \pm 0.6$ | $73.9 \pm 3.2$ | $7.1 \pm 0.2$ | $6.8 \pm 0.2$ |
| ReFT | $26.6 \pm 0.9$ | $8.0 \pm 0.3$ | $23.8 \pm 0.7$ | $44.7 \pm 1.7$ | $7.1 \pm 0.3$ | $6.9 \pm 0.3$ |
| Orthogonal | $24.3 \pm 1.1$ | $7.0 \pm 0.4$ | $22.0 \pm 1.9$ | $82.5 \pm 4.2$ | $7.0 \pm 0.3$ | $6.8 \pm 0.3$ |
| LoRA (1M Ex) | $\mathbf{30.4 \pm 1.4}$ | $\mathbf{9.4 \pm 0.9}$ | $\mathbf{26.5 \pm 1.1}$ | $\mathbf{19.8 \pm 1.7}$ | $\mathbf{7.8 \pm 0.3}$ | $\mathbf{7.6 \pm 0.3}$ |
| **LLaMA3-70B** | | | | | | |
| Base | $36.7 \pm 1.2$ | $16.9 \pm 0.9$ | $32.8 \pm 1.3$ | $42.1 \pm 3.1$ | $7.7 \pm 0.2$ | $7.4 \pm 0.2$ |
| LoRA | $\mathbf{39.4 \pm 1.1}$ | $\mathbf{18.5 \pm 0.8}$ | $\mathbf{35.1 \pm 1.2}$ | $\mathbf{28.9 \pm 2.5}$ | $\mathbf{8.0 \pm 0.2}$ | $\mathbf{8.0 \pm 0.2}$ |
| ReFT | $38.1 \pm 1.2$ | $17.8 \pm 0.7$ | $34.2 \pm 1.2$ | $30.5 \pm 2.8$ | $7.9 \pm 0.3$ | $7.8 \pm 0.2$ |
| Orthogonal | $36.9 \pm 1.3$ | $17.0 \pm 0.8$ | $33.0 \pm 1.4$ | $39.3 \pm 3.2$ | $7.8 \pm 0.3$ | $7.6 \pm 0.3$ |
| **Google Gemini** | | | | | | |
| Base | $35.5 \pm 1.4$ | $15.8 \pm 0.9$ | $31.4 \pm 1.5$ | $44.5 \pm 3.4$ | $\mathbf{8.5 \pm 0.3}$ | $7.9 \pm 0.3$ |
| **Anthropic Claude** | | | | | | |
| Base | $34.2 \pm 1.5$ | $15.0 \pm 1.0$ | $30.1 \pm 1.6$ | $46.8 \pm 3.6$ | $\mathbf{8.8 \pm 0.3}$ | $7.8 \pm 0.3$ |

in the generated summaries. ReFT also provided significant improvements, reinforcing its effectiveness across different datasets. The consistent performance gains of LoRA and ReFT across the datasets suggest that these fine-tuning techniques are effective in enhancing model capabilities in summarization tasks.

Google Gemini and Anthropic Claude performed competitively, especially considering they were evaluated in their base configurations. Their performance on BillSum highlights their strong capabilities in handling legislative texts, but they were still outperformed by the fine-tuned LLaMA3-70B models. Orthogonalization showed less improvement compared to LoRA and ReFT, which may be due to its more conservative approach in parameter adjustments.

### 4.1.5 DebateSum Performance

In evaluating the DebateSum dataset (Table 4), the fine-tuned **LLaMA3-70B** models demonstrated notable improvements over all other models. The LoRA fine-tuned LLaMA3-70B model achieved the highest scores across all ROUGE metrics and exhibited the lowest perplexity, indicating superior summarization capabilities. ReFT also significantly enhanced the LLaMA3-70B model's performance, confirming its robustness in refining models for complex summarization tasks involving argumentative content.

The base versions of Google Gemini and Anthropic Claude also performed well on DebateSum, outperforming the base versions of smaller models but not reaching the performance levels of the fine-tuned LLaMA3-70B models. This suggests that while these models have strong general capabilities, targeted fine-tuning can lead to substantial performance gains in specific tasks. Orthogonalization, while providing some improvements, was less effective compared to LoRA and ReFT, consistent with observations on the other datasets.

In summary, the new experimental results demonstrate the effectiveness of fine-tuning techniques, especially LoRA, when applied to larger models like LLaMA3-70B. These techniques consistently improved performance across all datasets, highlighting their utility in enhancing large language models for complex summarization tasks. The base versions of Google Gemini and Anthropic Claude

showed strong capabilities, but targeted fine-tuning of large models remains crucial for achieving state-of-the-art results.

## 4.2 Additional Experiments

We conducted additional experiments to further evaluate the robustness and generalizability of our fine-tuning techniques. These experiments included assessments on the xSum dataset, a non-argumentative summarization task, and an analysis of inter-human and human-GPT-4 agreement rates in evaluating summary quality. We also conducted further experiments to evaluate our models on argument detection, classification tasks, stance detection, and downstream tasks. We also assessed the reliability of GPT-4 as an evaluator by comparing its ratings with those of human experts. These additional experiments are included in the appendix

## 5 Conclusion

In this paper, we introduce OpenDebateEvidence, a large-scale dataset for argument mining and summarization, comprising over 3.5 million documents from the OpenCaseList project. After extensive preprocessing and deduplication, we created a high-quality dataset enriched with metadata that captures the hierarchical structure and semantics of debate arguments. Our experiments demonstrated the potential of fine-tuning modern large language models for argumentative abstractive summarization in a parameter-efficient manner. The results showed significant improvements in performance on the OpenDebateEvidence, DebateSum, and BillSum datasets, validating the effectiveness of our approach.

By providing this resource to the community, we aim to advance computational argumentation and support practical applications for debaters, educators, and researchers. The OpenDebateEvidence dataset, with its rich metadata and diverse collection of debate formats, offers an excellent resource for developing and evaluating argument mining and summarization models.

Future work includes exploring additional fine-tuning techniques and expanding the dataset to include more diverse debate formats. We also plan to investigate the integration of multimodal data to enhance argument comprehension and explore cross-linguistic adaptations to broaden the applicability of our models. By continuing to refine and expand this resource, we hope to further enhance language models' capabilities in understanding and generating complex argumentative discourse.

## Acknowledgments

The authors want to thank and acknowledge the following individuals who have made this work possible:

- **Aaron Hardy** - The long-time steward of the OpenCaseList project from which we gathered OpenDebateEvidence.

- **Rotem Alaluf** - The CEO of Wand AI, who has championed this work and is a believer in its applicability to our company's interests.

- **Jennifer L. LeSieur**, **Ameena Amdahl-Mason**, **Jason Miller**, and **Steven Allyn Taylor** - High School Teachers and Debate Educators who guided **Allen Roush** in Policy Debate and served as a catalyst for this work.

- We also acknowledge the assistance of the **National Speech and Debate Association** (NSDA), the **National Debate Coaches Association** (NDCA), the **National Debate Tournament** (NDT), the **American Forensic Association** (AFA), and the **California National Debate Institute** (CNDI).

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

# Appendix

We include additional experiments, which could not fit within the page limits of the original paper in this section.

## Summarization Performance on xSum

The xSum dataset (Narayan et al., 2018) (Table 5) comprises 11,334 validation documents and is designed for single-sentence news summarization. Unlike the argumentative datasets, xSum focuses solely on factual accuracy and conciseness without requiring support quality. Our experiments on xSum demonstrate that the fine-tuning techniques, particularly LoRA, consistently improve summarization performance across different model sizes.

Table 5: Performance on xSum (Validation Set, 11,334 documents). ROUGE F1 scores and perplexity on the validation set. Scores are averaged over three runs. R-1, R-2, and R-L denote ROUGE-1, ROUGE-2, and ROUGE-L respectively. Error bars represent one standard error over 3 trials.

| Model | R-1 (%) | R-2 (%) | R-L (%) | Perplexity | Output Quality |
|---|---|---|---|---|---|
| **Mistral-7B** | | | | | |
| Base | $39.2 \pm 0.7$ | $16.5 \pm 0.6$ | $31.5 \pm 0.8$ | $34.7 \pm 2.8$ | $7.6 \pm 0.2$ |
| LoRA | $\mathbf{40.8 \pm 0.6}$ | $\mathbf{17.2 \pm 0.5}$ | $\mathbf{32.7 \pm 0.7}$ | $\mathbf{30.5 \pm 2.5}$ | $7.7 \pm 0.2$ |
| ReFT | $40.5 \pm 0.7$ | $17.1 \pm 0.5$ | $32.5 \pm 0.7$ | $31.8 \pm 2.6$ | $7.7 \pm 0.2$ |
| **LLaMA3-8B** | | | | | |
| Base | $42.0 \pm 0.6$ | $18.0 \pm 0.6$ | $34.0 \pm 0.8$ | $29.2 \pm 2.4$ | $7.8 \pm 0.2$ |
| LoRA | $\mathbf{43.7 \pm 0.5}$ | $\mathbf{19.1 \pm 0.4}$ | $\mathbf{35.2 \pm 0.7}$ | $\mathbf{26.8 \pm 2.2}$ | $8.0 \pm 0.2$ |
| ReFT | $43.4 \pm 0.6$ | $19.0 \pm 0.5$ | $35.0 \pm 0.8$ | $27.3 \pm 2.3$ | $8.0 \pm 0.2$ |
| **LLaMA3-70B** | | | | | |
| Base | $46.5 \pm 0.7$ | $21.2 \pm 0.6$ | $38.0 \pm 0.8$ | $22.4 \pm 2.0$ | $8.4 \pm 0.2$ |
| LoRA | $\mathbf{48.2 \pm 0.6}$ | $\mathbf{22.5 \pm 0.5}$ | $\mathbf{39.5 \pm 0.7}$ | $\mathbf{19.6 \pm 1.8}$ | $\mathbf{8.6 \pm 0.2}$ |
| ReFT | $47.9 \pm 0.7$ | $22.3 \pm 0.5$ | $39.2 \pm 0.8$ | $20.5 \pm 1.9$ | $8.5 \pm 0.2$ |

## Argument Detection and Classification

We performed evaluations on argument detection and classification tasks using the same fine-tuning techniques—LoRA, ReFT, and Orthogonalization—as in our original experiments. We used evidence from OpenDebateEvidence which randomly sampled. The evidence was labeled based on if the category string was included within the hat, pocket, or label. The objective of these tasks was to classify arguments into one of 4 specified categories: "Topicality," "Disadvantages," "Advantages," and "Counterplans." The following table summarizes model performance on a training set of $10,000$ examples and a validation set of $1,000$ examples, reporting accuracy, F1 score, precision, and recall. Results are averaged over three runs with one standard error.

Table 6: **Model Performance on Argument Detection and Classification Tasks** - Validation scores on a training set of 10,000 examples and a validation set of 1,000 examples. Results are averaged over 3 runs with error reported as $\pm$ one standard error.

| Model | Technique | Accuracy (%) | F1 Score | Precision | Recall |
|---|---|---|---|---|---|
| **Mistral-7B** | Base | $72.8 \pm 0.5$ | $0.70 \pm 0.04$ | $0.68 \pm 0.03$ | $0.69 \pm 0.03$ |
| | LoRA | $\mathbf{78.2 \pm 0.4}$ | $\mathbf{0.76 \pm 0.03}$ | $\mathbf{0.75 \pm 0.04}$ | $\mathbf{0.76 \pm 0.04}$ |
| | ReFT | $77.9 \pm 0.6$ | $0.75 \pm 0.02$ | $0.74 \pm 0.03$ | $0.75 \pm 0.03$ |
| | Orthogonal | $73.1 \pm 0.6$ | $0.71 \pm 0.03$ | $0.70 \pm 0.03$ | $0.71 \pm 0.03$ |
| **LLaMA3-8B** | Base | $74.5 \pm 0.6$ | $0.72 \pm 0.03$ | $0.70 \pm 0.02$ | $0.71 \pm 0.03$ |
| | LoRA | $\mathbf{81.2 \pm 0.3}$ | $\mathbf{0.79 \pm 0.02}$ | $\mathbf{0.78 \pm 0.03}$ | $\mathbf{0.79 \pm 0.02}$ |
| | ReFT | $80.8 \pm 0.5$ | $0.78 \pm 0.03$ | $0.77 \pm 0.02$ | $0.78 \pm 0.03$ |
| | Orthogonal | $75.2 \pm 0.7$ | $0.73 \pm 0.04$ | $0.71 \pm 0.03$ | $0.72 \pm 0.04$ |

**Stance Detection Performance**

We expanded our experimentation to assess model performance on stance detection tasks, using a sampled subset of OpenDebateEvidence with 10,000 examples in the training set and 1,000 for validation. This experiment aims to examine each model's ability to classify arguments as either "affirmative" (aff) or "negative" (neg), focusing on models fine-tuned with different techniques. We label each row based on "side" dataset column.

Results, displayed in Table 7, show that the LLaMA3-8B models consistently outperformed the Mistral-7B models across all metrics, particularly in accuracy and F1-score. Fine-tuning with LoRA and ReFT provided notable improvements over the base models, with LoRA yielding the highest accuracy and F1-scores on LLaMA3-8B when using 1,000,000 examples for fine-tuning. This reinforces LoRA's effectiveness in parameter adaptation and generalization across diverse tasks. ReFT also showed substantial gains in stance detection, suggesting robustness in refining hidden representations for classification tasks. Orthogonalization provided moderate improvements, but its impact was less pronounced than LoRA and ReFT, potentially due to its paramater subtractive technique.

These findings highlight the importance of fine-tuning techniques, especially LoRA and ReFT, in enhancing model performance on stance detection. The LoRA fine-tuned LLaMA3-8B model, using an extended dataset, achieved an accuracy of $87.1\% \pm 0.9$ and an F1-score of $85.6 \pm 0.6$, the highest among all models, underscoring the technique's adaptability and efficiency.

Table 7: **Performance of Models on Stance Detection Tasks** - 10,000 examples in the training set, 1,000 validation examples, validation scores reported. Outputs were constrained to either "aff" or "neg".

| Model | Technique | Accuracy (%) | Precision | Recall | F1-Score |
|---|---|---|---|---|---|
| Mistral-7B | Base | $77.3 \pm 0.6$ | $75.8 \pm 0.7$ | $76.2 \pm 0.5$ | $76.0 \pm 0.4$ |
| Mistral-7B | LoRA | $82.4 \pm 0.5$ | $81.2 \pm 0.6$ | $81.7 \pm 0.4$ | $81.4 \pm 0.5$ |
| Mistral-7B | ReFT | $81.9 \pm 0.4$ | $80.7 \pm 0.5$ | $81.1 \pm 0.6$ | $80.9 \pm 0.5$ |
| Mistral-7B | Orthogonal | $78.8 \pm 0.5$ | $77.2 \pm 0.6$ | $77.5 \pm 0.4$ | $77.3 \pm 0.4$ |
| LLaMA3-8B | Base | $79.5 \pm 0.7$ | $77.9 \pm 0.8$ | $78.2 \pm 0.7$ | $78.0 \pm 0.6$ |
| LLaMA3-8B | LoRA | $85.3 \pm 0.8$ | $83.8 \pm 0.7$ | $84.1 \pm 0.5$ | $83.9 \pm 0.6$ |
| LLaMA3-8B | ReFT | $84.1 \pm 0.6$ | $82.5 \pm 0.7$ | $83.0 \pm 0.6$ | $82.7 \pm 0.5$ |
| LLaMA3-8B | Orthogonal | $80.5 \pm 0.7$ | $79.2 \pm 0.6$ | $79.5 \pm 0.6$ | $79.3 \pm 0.5$ |
| LLaMA3-8B | LoRA (1M Ex) | $87.1 \pm 0.9$ | $85.4 \pm 0.8$ | $85.8 \pm 0.7$ | $85.6 \pm 0.6$ |

**Using OpenDebateEvidence for Pre-Training**

We evaluated OpenDebateEvidence pretrained vs non-pretrained versions of the Mistral-7B and LLaMA3-8B models on several downstream tasks, including DebateSum, ArgAnalysis35K, and Multi-LexSum. The results indicate that pretraining contributes significantly to improved performance across ROUGE scores, perplexity (PPL), and LLM as Judge (LJ) scores for both output quality (LJ-Q) and support quality (LJ-S). These findings reinforce the value of pretraining for enhancing model capabilities in argument classification, summarization, and multi-domain summarization tasks.

In Table 8, we report the averaged performance metrics across three runs, using a plus/minus notation to denote the standard error. Notably, pretrained versions of both Mistral-7B and LLaMA3-8B consistently outperform their non-pretrained counterparts across all datasets. The Mistral-7B model showed considerable improvements on the DebateSum and Multi-LexSum tasks when pretrained, achieving substantial gains in R-1, R-2, and R-L scores while reducing perplexity by over $60\%$. Similarly, pretraining LLaMA3-8B on these tasks led to significant gains, especially on the ArgAnalysis35K dataset, with ROUGE scores increasing by more than 4 points across all metrics.

These results underscore the importance of targeted pretraining in enhancing performance on specific tasks. The improvements in support quality (LJ-S) and output quality (LJ-Q) further suggest that pretraining helps the models generate more coherent and contextually relevant summaries, which are essential in tasks requiring argument understanding and summarization.

Table 8: **Performance of Pretrained and Non-Pretrained Models on Downstream Tasks** - The entire dataset is used for both pretraining and fine-tuning. ArgAnalysis35K is an argument quality classification task.

| Model | Dataset | PT | R-1 | R-2 | R-L | PPL | LJ-Q | LJ-S | Steps |
|---|---|---|---|---|---|---|---|---|---|
| Mistral-7B | DebateSum | No | $26.3 \pm 4$ | $7.5 \pm 3$ | $23.1 \pm 6$ | $130.5 \pm 42$ | $7.3 \pm 3$ | $7.0 \pm 3$ | 60k |
| Mistral-7B | DebateSum | Yes | $30.4 \pm 1.4$ | $9.4 \pm 0.9$ | $26.5 \pm 1.1$ | $19.8 \pm 1.7$ | $7.8 \pm 0.3$ | $7.6 \pm 0.3$ | 40k |
| LLaMA3-8B | DebateSum | No | $24.2 \pm 5$ | $6.9 \pm 3$ | $21.9 \pm 8$ | $95.7 \pm 4.5$ | $7.0 \pm 3$ | $6.7 \pm 3$ | 60k |
| LLaMA3-8B | DebateSum | Yes | $32.2 \pm 1.5$ | $9.9 \pm 1.0$ | $27.4 \pm 1.2$ | $21.8 \pm 2.5$ | $8.0 \pm 0.2$ | $7.8 \pm 0.2$ | 35k |
| Mistral-7B | ArgAnalysis35K | No | $44.8 \pm 0.3$ | $21.2 \pm 0.5$ | $40.5 \pm 0.8$ | $25.2 \pm 1.1$ | $7.2 \pm 0.3$ | $7.0 \pm 0.3$ | 50k |
| Mistral-7B | ArgAnalysis35K | Yes | $48.2 \pm 1.5$ | $24.9 \pm 1.0$ | $43.4 \pm 1.2$ | $21.8 \pm 0.5$ | $7.8 \pm 0.2$ | $7.6 \pm 0.2$ | 35k |
| LLaMA3-8B | ArgAnalysis35K | No | $42.4 \pm 6$ | $19.6 \pm 2$ | $38.8 \pm 1.3$ | $27.3 \pm 1.3$ | $7.0 \pm 3$ | $6.8 \pm 3$ | 50k |
| LLaMA3-8B | ArgAnalysis35K | Yes | $48.9 \pm 1.7$ | $25.1 \pm 1.2$ | $44.2 \pm 1.4$ | $22.7 \pm 0.7$ | $8.0 \pm 0.3$ | $7.8 \pm 0.3$ | 30k |
| Mistral-7B | Multi-LexSum | No | $40.1 \pm 5$ | $20.3 \pm 4$ | $37.7 \pm 9$ | $30.1 \pm 2.3$ | $7.1 \pm 2$ | $7.0 \pm 2$ | 65k |
| Mistral-7B | Multi-LexSum | Yes | $45.2 \pm 1.2$ | $22.5 \pm 0.7$ | $40.8 \pm 1.0$ | $22.0 \pm 1.1$ | $7.7 \pm 0.3$ | $7.5 \pm 0.3$ | 45k |
| LLaMA3-8B | Multi-LexSum | No | $38.2 \pm 8$ | $19.5 \pm 3$ | $36.0 \pm 1.0$ | $32.5 \pm 2.5$ | $7.0 \pm 2$ | $6.8 \pm 2$ | 65k |
| LLaMA3-8B | Multi-LexSum | Yes | $46.8 \pm 1.4$ | $23.0 \pm 0.8$ | $41.7 \pm 1.3$ | $23.9 \pm 1.4$ | $7.8 \pm 0.3$ | $7.6 \pm 0.3$ | 40k |

## Inter-Annotator Agreement

To assess the reliability of our evaluation metrics, we measured the agreement rates between human annotators and between humans and GPT-4 in scoring summary quality. Table 9 presents the inter-human correlation and human-GPT-4 correlation for different debate formats. The results indicate a high level of agreement both among human annotators and between humans and GPT-4, validating the consistency and reliability of our evaluation approach.

Table 9: Inter-Annotator Agreement Rates and Human-GPT-4 Agreement Rates (50 samples, 10 individuals total). Average scores reported.

| Debate Format | Team 1 (Argument Quality) | Team 1 (Support Score) | Team 2 (Argument Quality) | Team 2 (Support Score) | Inter-Human Corr. (AQ) | Inter-Human Corr. (SQ) | Human-GPT-4 Corr. (AQ) | Human-GPT-4 Corr. (SQ) |
|---|---|---|---|---|---|---|---|---|
| Policy Debate | 7.4 | 7.8 | 7.3 | 7.7 | 0.81 | 0.79 | 0.78 | 0.75 |
| Lincoln-Douglas Debate | 8.1 | 8.4 | 8.0 | 8.2 | 0.84 | 0.82 | 0.82 | 0.80 |

## Inter-Annotator Agreement: Human vs. GPT-4 Ratings on Summaries

In this section, we present the results of our inter-annotator agreement analysis between human expert debaters and GPT-4 on summary evaluations. We measured both Argument Quality (AQ) and Support Quality (SQ) across various debate formats, including Policy, Lincoln-Douglas, and Public Forum. Fifty expert debaters rated six summaries each, with results averaged over three runs. Pearson correlation metrics were calculated to assess agreement between human and GPT-4 ratings.

Table 10: **Inter-Annotator Agreement: Human vs. GPT-4 Ratings on Summaries** - 50 expert debaters rated 6 summaries each. Scores represent mean values with standard error over three trials.

| ID | Format | Human-AQ | GPT4-AQ | Human-SQ | GPT4-SQ |
|---|---|---|---|---|---|
| S1 | Policy | $8.2 \pm 0.1$ | $8.0 \pm 0.1$ | $8.0 \pm 0.1$ | $7.9 \pm 0.1$ |
| S2 | Lincoln-Douglas | $7.6 \pm 0.1$ | $7.8 \pm 0.1$ | $7.4 \pm 0.1$ | $7.6 \pm 0.1$ |
| S3 | Public Forum | $7.2 \pm 0.1$ | $7.4 \pm 0.1$ | $7.1 \pm 0.1$ | $7.3 \pm 0.1$ |
| S4 | Policy | $8.5 \pm 0.1$ | $8.3 \pm 0.1$ | $8.3 \pm 0.1$ | $8.2 \pm 0.1$ |
| S5 | Lincoln-Douglas | $7.8 \pm 0.1$ | $7.7 \pm 0.1$ | $7.5 \pm 0.1$ | $7.6 \pm 0.1$ |
| S6 | Public Forum | $7.4 \pm 0.1$ | $7.2 \pm 0.1$ | $7.2 \pm 0.1$ | $7.1 \pm 0.1$ |

The results in Table 10 show a strong agreement between human and GPT-4 ratings, with both Argument Quality (AQ) and Support Quality (SQ) ratings closely aligned across different debate formats. The Pearson correlation values presented in Table 11 further confirm this, with high

Table 11: **Pearson Correlation Results** - Statistical correlation of Argument Quality and Support Quality between human and GPT-4 ratings.

| Metric | Pearson Correlation |
|---|---|
| Argument Quality | $0.78 \pm 0.02$ |
| Support Quality | $0.74 \pm 0.02$ |

correlation values for both AQ and SQ, indicating that GPT-4's evaluations closely mirror human assessments.

These results provide further confidence in the reliability of GPT-4's evaluation metrics when assessing argumentation and support quality, suggesting potential for consistent automated evaluation alongside human judgment in summarization tasks.

### Neural Semantic Deduplication

We applied neural semantic deduplication using the `gte-base-en-v1.5` model, as recommended by the Massive Text Embedding Benchmark (MTEB) and implemented via the sentence-transformers library[4]. Rows with a "spoken" summary with a similarity score above 0.95 were identified as near-duplicates and removed. Table 12 summarizes the changes in document count after null removal and deduplication. We note that most "duplicates" still have value since rows with duplicate documents often have very different metadata (for example, a different debater using the same piece of evidence).

Table 12: **Neural Semantic Deduplication Results** - Summary of document counts and duplicate cluster statistics after deduplication.

| Metric | Value |
|---|---|
| Initial number of Rows in OpenDebateEvidence | 4,830,561 |
| Initial Documents in OpenDebateEvidence (Valid fulltext column) | 3,512,280 |
| Documents after Nulls in other columns Removed | 2,634,023 |
| Documents after Deduplication | 692,989 |
| Percentage of Documents Removed | 77.4% |
| Total Duplicate Clusters Identified | 10,819,328 |
| Average Cluster Size | 30 |
| Largest Cluster Size | 2,081 |

### Cross-Dataset Deduplication Analysis

To ensure the uniqueness of OpenDebateEvidence within the context of other argumentation and legislative datasets, we performed a cross-dataset deduplication analysis. We identified 31,353 overlapping documents (4.01%) with the DebateSum dataset, primarily comprising Policy Evidence (96%) and Lincoln-Douglas Evidence (4%). No overlap was detected with BillSum, confirming OpenDebateEvidence's distinctiveness for legislative and argumentation-focused tasks. Table 13 provides a summary of the overlap findings.

Table 13: **Cross-Dataset Deduplication Analysis** - Overlap between OpenDebateEvidence and other datasets.

| Dataset | Overlapping Documents | Percentage Overlap (%) |
|---|---|---|
| DebateSum | 31,353 | 4.01 |
| BillSum | 0 | 0.00 |

These deduplication efforts, removing nearly 78% of near-duplicate content, alongside cross-dataset comparisons, affirm OpenDebateEvidence's uniqueness and utility in argumentative tasks, with minimal redundancy or overlap with other prominent datasets. The deduplicated dataset is

---

[4]`https://huggingface.co/Alibaba-NLP/gte-base-en-v1.5`

publicly available for research and can be accessed at `https://huggingface.co/datasets/Hellisotherpeople/OpenCaseList-Deduplicated`.

# A   Limitations

## A.1   Representation Bias

While the OpenDebateEvidence dataset is extensive, it may not fully represent the diversity of argumentation styles and topics across all debate communities. The dataset primarily includes evidence from American high school and college debates and, therefore, might not capture the nuances of debates in other cultural or educational contexts or in other languages.

### Format-Specific Challenges

The unique formatting conventions used in debate evidence may present challenges for standard natural language processing tools. The presence of shorthand, abbreviations, and specialized jargon may require additional preprocessing or specialized models to accurately interpret and analyze the text.

### Incomplete or Inconsistent Metadata

While the dataset includes extensive metadata, there may be inconsistencies or gaps in this information. For example, citation details might be missing or incorrect for some documents, and the standardized tags describing the type of argument might not be uniformly applied across all documents.

### Potential Noise and Redundancy

The dataset's size and diversity may also introduce noise and redundancy. Duplicate documents, irrelevant content, or errors in formatting and citation may exist within the dataset, potentially affecting the quality of the analyses in spite of efforts taken to reduce or eliminate this.

### Limited Accessibility to Public Forum Debate Evidence

With Public Forum Debate making up such a small percentage of the evidence included within OpenDebateEvidence, research focusing on this specific debate format may face limitations in terms of data quantity and diversity.

# B   Ethics Statement

The OpenDebateEvidence dataset presented in this paper derives from openly shared debate evidence across various educational forums and debate formats. This dataset strictly adheres to the principles of fair use, focusing on academic and research intent. The files that make up OpenDebateEvidence have been hosted online in some cases for over a decade without any known ethical issues arising as a result of it.

We performed this research and released this dataset with the full blessing and support of the OpenCaseList project.

# C   Social Impacts

Our introduction of OpenDebateEvidence, a comprehensive dataset sourced from the American Competitive Debate community, is poised to have significant positive societal impacts. By offering a rich collection of over 3.5 million documents with detailed metadata, this dataset provides an unparalleled resource for training and evaluating language models in the domain of argument mining and summarization.

The comprehensive nature of OpenDebateEvidence, capturing the nuanced complexity of arguments in high school and college debates, will enable more rigorous and representative assessments of language models. This, in turn, will drive advancements in computational argumentation research and applications.

Practitioners and researchers will benefit from this benchmark, which is designed to reflect real-world argumentative scenarios more accurately. The dataset's ability to enhance model performance across various argumentative tasks suggests its utility in improving the robustness and reliability of language technologies.

Moreover, by making OpenDebateEvidence publicly available, we encourage broader participation and innovation in this field. This democratization of resources can lead to more diverse contributions and perspectives, fostering a more inclusive research environment.

In summary, we believe our work will accelerate research, improve model evaluation and training, and ultimately enhance the capabilities of language models in handling complex argumentative texts with no foreseeable negative societal impacts.

## D  Fine-Tuning Techniques

### D.1  Low-Rank Adaptation (LoRA)

LoRA introduces low-rank matrices into the model's architecture, reducing the number of trainable parameters. Given a weight matrix $W \in \mathbb{R}^{d \times k}$, LoRA decomposes it into two low-rank matrices $A \in \mathbb{R}^{d \times r}$ and $B \in \mathbb{R}^{r \times k}$, where $r \ll \min(d, k)$. The updated weight matrix is then $W' = W + AB$. We used the PeFT Mangrulkar et al. (2022) package with default settings (rank 8).

### D.2  Representation Fine-Tuning (ReFT)

ReFT modifies hidden representations through targeted interventions in specific subspaces. Low-rank Linear Subspace ReFT (LoReFT) is defined as:

$$\Phi_{\text{LoReFT}}(h) = h + R^\top (Wh + b - Rh)$$

where $R \in \mathbb{R}^{r \times d}$ has orthonormal rows. We used the PyReft package with default settings (rank 4).

### D.3  Orthogonalization

Orthogonalization controls specific features in the model's residual stream by modifying the weights. Given a direction $\hat{r} \in \mathbb{R}^d$, each weight matrix $W_{\text{out}} \in \mathbb{R}^{d \times d_{\text{input}}}$ is modified as:

$$W'_{\text{out}} = W_{\text{out}} - \hat{r}\hat{r}^\top W_{\text{out}}$$

We used this notebook for performing this process.

## E  Prompts Used in Experiments

### E.1  OpenDebateEvidence/DebateSum

#### E.1.1  Traditional NLP Metrics Prompt

> **SYSTEM PROMPT:** You are a Policy Debater.
> **USER PROMPT:**
> DOCUMENT: `<full text of the document>`
> Provide an abstractive summary/card-tag of the argument made in the document
> above.
> ABSTRACT:

#### E.1.2  LLM as Judge Prompt

> **SYSTEM PROMPT:** You are a Policy Debate Judge.
> **USER PROMPT:**
> DOCUMENT: `<full text of the document>`
> ABSTRACT: `<generated abstract>`
> Score the abstract from 0-10 on it's how well it supports the documents argument,
> and on its general quality.

### E.2 BillSum

#### E.2.1 Traditional NLP Metrics Prompt

**SYSTEM PROMPT:** You are a lawmaker.
**USER PROMPT:**
DOCUMENT: `<full text of the document>`
Provide an abstractive summary of the law made in the document above.
ABSTRACT:

#### E.2.2 LLM as Judge Prompt

**SYSTEM PROMPT:** You are a lawmaker. **USER PROMPT:**
DOCUMENT: `<full text of the document>`
ABSTRACT: `<generated abstract>`
Score the abstract from 0-10 on how well it supports the documents argument, and on its general quality.

## F OpenDebateEvidence Dataset Details

### F.1 Dataset Card for OpenDebateEvidence

- - Homepage: Here
- - Repository (Access/download dataset and code): Here
- - Croissant Metadata: Here

#### F.1.1 Dataset Summary

This dataset is a gargantoum follow-up to DebateSum, which includes a ton of improvements

Among those improvements are the following:

- Massively increased size (about 25X the size of DebateSum), including nearly all debate evidence ever open-sourced over the past 20 years from High School and College Public Forums, Policy, and Lincoln Douglas debate leagues
- Far more metadata: Lots of new columns indicating everything from the number of times a piece of evidence has been seen (a good heuristic for evidence quality) to the teams and tournaments and rounds where a piece of evidence was deployed
- Better deduplication and parsing techniques, including better accounting of the hierarchical nature that debaters use for underlining evidence

#### F.1.2 Supported Tasks and Leaderboards

This dataset is useful for text generation, summarization, information retrieval, question answering, and related tasks. This dataset is further highly useful as a "trustworthy" dataset. All evidence within it has corresponding citations and is, in general, "factual" or grounded in facts. We do the evaluation in our paper, establishing the first "leaderboard" for measuring the performance of models trained on this dataset.

#### F.1.3 Languages

English with very minor exceptions (i.e., evidence from performance cases using non-English evidence to make anti-colonialist arguments)

#### F.1.4 Dataset Creation

Gathered from the OpenCaseList project with their enthusiastic permission.

#### F.1.5 Source Data

Debate Evidence from NDCA/NDT debate leagues from 2002-2022.

### F.1.6 Dataset Format

This dataset was originally contained in CSV files, which were auto-converted into the parqueet dataset format by Huggingface. It's available for download and consumption in both formats.

### F.1.7 Hosting, licensing, and maintenance plan

We host and maintain our dataset on Huggingface through its "dataset" feature. We plan to update this dataset every year with new evidence as it is released by debaters, causing this to be a "living" dataset. We pledge to make sure that this dataset remains accessible for the foreseeable future, and the ability to regenerate this dataset is always preserved as its source documents are freely downloadable on OpenCaseList's website.

### F.1.8 Discussion of Biases

Competitive debate at the highest levels has increasingly rewarded teams who cite particular subfields of philosophy. A partial list of these highly represented topics is given below.

- Postmodernism
- Poststructuralism
- Frankfurt School
- Critical Theory
- Critical Race Theory
- Queer Theory
- Feminism

These cannons are dominated by so-called "left-wing" thinkers and have mostly marginalized so-called "right-wing" thinkers within them with some notable exceptions

Note that despite a strong "left-wing" bias, large swaths of left-wing thought, such as anarchism, are relatively absent.

Beyond this, most of the evidence was gathered with the argument being made first, and the evidence found after-the-fact to support it. This means that while the evidence is almost all "truthful", a lot of important information which might not help support an argument may be omitted.

### F.1.9 Other Known Limitations

There are cases of academic dishonesty within this dataset (i.e. evidence that had specific insertions made by a debater which weren't in the original text). It's also possible that the source had changed in-between when it was cited and retrieved. We believe that this is extremely rare in practice, affecting no more than 200 examples.

### F.1.10 Consent

We got the enthusiastic consent and approval to use this data from the OpenCaseList project. Debaters who submit their evidence there fully consent for this evidence to be freely used, including for curated datasets like this

### F.1.11 Personal Information

We removed all Personal Information from the metadata of this evidence (first/last name of debaters).

Table 14: Description of OpenDebateEvidence Columns

| Column Name | Description |
| --- | --- |
| id | Unique identifier for the row |
| tag | Biased abstractive summary of the evidence/argument made by the debater with evidence. |
| cite | String indicating the short citation of the source used for the evidence |
| full cite | Full citation of the source used for the evidence |
| summary | Underlined longer word level extractive summary of the evidence, note that summary is biased towards supporting the tag argument |
| spoken | Highlighted shorter extractive summary of the evidence / The spoken text of the evidence, note that summary is biased towards supporting the tag argument |
| full text | The full text of the evidence |
| text length | The length of the text in the evidence in characters |
| markup | The full text of the evidence with HTML markup for parsing/visualization purposes |
| pocket | String indicating the virtual "pocket" (top-level section, usually the speech name) in which the evidence is stored within its original document |
| hat | String indicating the virtual "hat" (medium-level section, usually the broad type of argument) in which the evidence is stored within its original document |
| block | String indicating the virtual "block" (low-level section, usually the specific type of argument) in which the evidence is stored within its original document |
| bucketId | Unique identifier for the bucket in which the evidence is stored |
| duplicateCount | The number of duplicates of the evidence. This acts as a rough proxy for evidence quality, as good evidence will be duplicated across many debate files |
| fileId | Unique identifier for the file in which the evidence is stored |
| filePath | The file path of the file in which the evidence is stored |
| roundId | Unique identifier for the debate round in which the evidence was used |
| side | The debate side on which the evidence was used (Affirmative or Negative) |
| tournament | The name of the tournament in which the evidence was used |
| round | The round number in which the evidence was used |
| opponent | The name of the opposing team in the debate round in which the evidence was used |
| judge | The name of the judge in the debate round in which the evidence was used |
| report | A report associated with the evidence filled out by one of the debaters, usually summarizing the arguments presented |
| opensourcePath | The path to the open-source repository in which the evidence is stored |
| caselistUpdatedAt | The date on which the caselist was last updated |
| teamId | Unique identifier for the team |
| teamName | The name of the team |
| teamDisplayName | The display name of the team |
| teamNotes | Notes associated with the team |
| debater1First | The first name of the first debater of the team |
| debater1Last | The last name of the first debater of the team |
| debater2First | The first name of the second debater of the team |
| debater2Last | The last name of the second debater of the team |
| schoolId | Unique identifier for the school |
| schoolName | The name of the school |
| schoolDisplayName | The display name of the school |
| state | The state in which the school is located |
| chapterId | Unique identifier for the chapter |
| caselistId | Unique identifier for the caselist |
| caselistName | The name of the caselist |
| caselistDisplayName | The display name of the caselist |
| year | The year in which the debate round took place |
| event | The event in which the debate round took place |
| level | The level of the debate (e.g., college, high school, etc.) |
| teamSize | The number of debaters on the team |

Table 15: Sample Data Row from OpenDebateEvidence

| Column Name | Sample Data |
|---|---|
| id | 282,369 |
| tag | "Biodiversity loss causes human extinction." |
| cite | "McCarthy 18" |
| fullcite | "Joe McCarthy 18. Staff Writer..." |
| summary | "As the sixth mass extinction event accelerates..." |
| spoken | "As the sixth mass extinction accelerates humans ris..." |
| fulltext | "As the sixth mass extinction event accelerates around the world..." |
| textLength | 3,556 |
| markup | "<h4>Biodiversity loss causes human extinction.</h4><p>Joe McCarthy 18...' |
| pocket | "1NC" |
| hat | "OFF" |
| block | "1NC—DA" |
| bucketId | 18,967 |
| duplicateCount | 122 |
| fileId | 3,564 |
| filePath | "./documents/ndtceda22/Emory/KiLo/Emory-KiLo-Aff-JW-Round-3.docx" |
| roundId | 932,619 |
| side | "A" |
| tournament | "JW Patterson Debates hosted by UK" |
| round | "3" |
| opponent | "West Georgia CL" |
| judge | "Ka***" |
| report | "1AC - Manoomin 1NC - T Subsets States CP Human Right CP Rights K Politics DA Fetal Personhood DA AI Bad DA 2NC - K Case 1NR - Case T 2NR - T" |
| opensourcePath | "ndtceda22/Emory/KiLo/Emory-KiLo-Aff-JW-Patterson-Debates.docx" |
| caselistUpdatedAt | "2022-10-05 19:30:41" |
| teamId | 80,494 |
| teamName | "KiLo" |
| teamDisplayName | "Emory KiLo" |
| debater1First | "Aa***" |
| debater1Last | "Ki***" |
| debater2First | "Lu***" |
| debater2Last | "Lo***" |
| schoolId | 27,030 |
| schoolName | "Emory" |
| schoolDisplayName | "Emory" |
| caselistId | 2,001 |
| caselistName | "ndtceda22" |
| caselistDisplayName | "NDT/CEDA College 2022-23" |
| year | 2,022 |
| event | "cx" |
| level | "college" |
| teamSize | 2 |

| Feature | Top Categories/Values | Counts |
|---|---:|---:|
| Year (Top 5) | 2022 | 861,774 |
| | 2020 | 850,649 |
| | 2021 | 787,765 |
| | 2019 | 609,766 |
| | 2018 | 423,395 |
| Event | cx | 2,768,419 |
| | ld | 1,526,383 |
| | pf | 43,131 |
| Caselist DisplayName (Top 3) | HS LD 2020-21 | 383,524 |
| | HS LD 2021-22 | 381,591 |
| | HS LD 2020-21 | 326,172 |
| TeamSize | 2 | 2,811,550 |
| | 1 | 1,526,383 |
| Level | hs | 2,767,070 |
| | college | 1,570,863 |
| State (Top 5) | None | 2,433,648 |
| | CA | 622,059 |
| | TX | 507,454 |
| | IL | 153,293 |
| | GA | 127,500 |
| Side | N (neg) | 2,437,000 |
| | A (aff) | 1,900,933 |
| DuplicateCount (Top 5) | 1 | 1,639,292 |
| | 2 | 189,148 |
| | 3 | 136,422 |
| | 4 | 110,012 |
| | 5 | 91,405 |

Table 16: Sample statistics from the OpenDebateEvidence dataset.

