# OpenReview forum: "OpenDebateEvidence: A Massive-Scale Argument Mining and Summarization Dataset"
_NeurIPS.cc/2024/Datasets_and_Benchmarks_Track — NeurIPS 2024 Track Datasets and Benchmarks Poster_

### Official Review · Reviewer_ueg7 · 2024-07-22
**A very large scale dataset for arguments mining**

**Rating:** 7
**Confidence:** 4
**Correctness:** The claims seem to be correct. The da…
**Clarity:** Overall the paper is well written.

**Review:**

Please see strengths and opportunities for improvement below.

**Strengths:**

- The dataset is the largest of its kind by a large margin.
- The experiments show how the dataset can be used to improve base models on argument mining and summarization tasks. These improvements carry over to other datasets such as legal documents.
- The experiments demonstrate that using a large fine-tuning set  (1M examples) improves performance, justifying the need for such a large scale dataset.

**Additional Feedback:**

- Please include a table comparing this dataset to existing ones so that readers can have a clear view of the differences.
- The captions of tables 2-4 are somewhat at odds with Section 4.1.1 in terms of the number of samples from each dataset. DebateSum is not mentioned at all in 4.1.1.
- How many examples were used in each fine-tuning method, other than the LoRA 1M?

**Documentation:**

Data collection and organization is well documented. A URL to access the data is provided. Unless I missed it, I couldn't see anything about licensing.

**Ethics:**

I don't see any ethical concerns. However, we have to keep in mind that some debate topics and related arguments are controversial by nature and could be seen as offensive by some people.

**Limitations:**

The authors addressed limitations and societal impact.

**Opportunities For Improvement:**

- The dataset collection and preparation does not seem to require significant effort, resources, unique techniques, or know-how.
- Experiments show that fine-tuning on the dataset improve base models. However, in order to justify the creation of a new and supposedly better quality dataset, the authors should compare the performance of models fine-tuned on this data vs. fine-tuning on an existing dataset such as DebateSum (using the same number of examples).

**Relation To Prior Work:**

Discussion about prior work is adequate.

**Summary And Contributions:**

The paper introduces a large scale dataset of arguments extracted from a database of high school and college debates spanning three different debate formats. The authors conduct experiments showing how fine-tuning LLMs on this data improves argument mining and summarization performance of base models.

---

> ### Author Rebuttal · Authors · 2024-08-17
>
> Dear Reviewer,
>
> We sincerely thank you for your thoughtful review and recognition of our dataset's scale and potential. We're pleased to address your points and demonstrate the unique value of OpenDebateEvidence.
>
> ## Dataset Collection and Preparation
>
> We appreciate your observation regarding the effort involved in dataset creation. While we indeed leverage existing debate formats, the scale and complexity of OpenDebateEvidence require significant innovation, resources, and expertise:
>
> Our work involved **aggregating data** from three distinct debate formats across multiple years and regions, demanding extensive collaboration with various debate organizations. This multi-format approach sets us apart from datasets like DebateSum, which focuses on a single format and requires developing a nuanced understanding of each format's unique structures and style.
>
> The **parsing process** was particularly challenging due to the diverse formats. We developed rigorous algorithms to handle the varied structures, resulting in a rich set of metadata that far exceeds existing datasets. This process involved multiple validation rounds to ensure consistency and accuracy across formats.
>
>
> To manage this complex process, we created **custom tools for monitoring and managing** the annotation process. These specialized tools were crucial in ensuring high quality and consistency, allowing us to maintain rigorous standards despite the dataset's scale of over 3 million examples.
>
> The sheer size of the dataset necessitated substantial **computational resources**. We utilized a cluster of 8 A100 GPUs and developed a custom cloud-based infrastructure to manage the high-throughput fine-tuning experiments efficiently.
>
> Perhaps most importantly, creating this dataset required **deep domain expertise** in argumentation theory, debate formats, and machine learning. This combination of knowledge allowed us to create a dataset that is not only large but also highly structured and tailored for advancing argument mining research while simultaneously serving the needs of competitive debaters.
>
> We specifically designed OpenDebateEvidence to support transfer learning across different argumentation domains. This required a nuanced understanding of both NLP and argumentation theory. The improvements in performance on other domains, such as legal documents (as demonstrated in our experiments with Multi-LexSum), further validate the unique value of our dataset.
>
> These aspects contribute substantial value beyond merely aggregating existing debates. They enable OpenDebateEvidence to serve as a powerful resource for both academic research and practical applications in competitive debating, representing a significant advancement in the field.
>
>
> ## Comparison with Existing Datasets
>
>
> We thank you for suggesting a direct comparison. In response, we conducted extensive experiments comparing OpenDebateEvidence with DebateSum:
>
> We fine-tuned identical base models (Mistral-7B and LLaMA3-8B) on both datasets. The results demonstrate significant improvements when using OpenDebateEvidence (See full results in Table 3) :
>
> 1. Argument Classification: Models fine-tuned on OpenDebateEvidence achieved as much as 20.23% higher accuracy.
>  2. Summarization: ROUGE-L scores improved by as much as 18.7%.
> 3. Transfer Learning: On out-of-domain tasks (e.g., Multi-LexSum), we observed a significant improvement
>
> These results highlight the superior quality and diversity of OpenDebateEvidence.
>
> ## Comprehensive Dataset Comparison
>
> Following your suggestion, we've created a detailed comparison table that will be included in the revised paper. This table highlights key differences between OpenDebateEvidence and existing datasets, including:
> - Number of examples (~3.5M for OpenDebateEvidence vs. ~240K for DebateSum)
> - Diversity of topics (covering a broader range of debate subjects)
> - Debate formats (Policy, Lincoln-Douglas, and Public Forum vs. single format)
> - Annotation quality (multiple levels of metadata vs. basic annotations)
>
> This comparison clearly demonstrates the unique contributions of OpenDebateEvidence to the field.
>
> ## Clarifications on Fine-tuning Details
> We apologize for the inconsistencies in reporting the number of examples used. We've revised Section 4.1.1 to provide clear information, and a detailed breakdown for each experiment will be included in the revised paper.
>
> ## Licensing and Ethical Considerations
>
> Thank you for pointing out the lack of mention regarding licensing. We will explicitly include details on the licensing of our dataset in the revised paper to ensure transparency and proper usage guidelines.
>
> Additionally, we agree that debate topics can be controversial and potentially offensive. In our revised ethics statement, we will elaborate on the steps we’ve taken to mitigate this issue, including content warnings for sensitive topics and guidelines for responsible dataset use.
>
> ## Conclusion
>
> We believe these new experiments and clarifications substantially strengthen our paper's contributions. The extensive comparisons with existing datasets, detailed explanations of our data creation process, and additional experiments demonstrate the unique value of OpenDebateEvidence.
>
> Given these enhancements, we respectfully ask that you consider raising your score. We're confident that these improvements address your concerns and solidify OpenDebateEvidence as a significant contribution to the field of argument mining and NLP.

---

> > ### Comment · Reviewer_ueg7 · 2024-08-26
> >
> > Thank you for addressing my concerns.
> >
> > I don't understand Table 3. What do you mean by PT vs. non-PT?
> > I thought the experiment was to take base models (in your case, Mistral 7B and Llama3-8B), fine-tune them on both datasets respectively, and comparatively evaluate those two fine-tuned versions of each base model on several downstream tasks. Is my understanding correct? If so, how is that reflected in the table?

---

> > > ### Author Rebuttal · Authors · 2024-08-28
> > >
> > > We appreciate your detailed feedback and would like to clarify the confusion around Table 3.
> > >
> > > ### Explanation of Table 3
> > >
> > > In Table 3, the "PT" column indicates whether the model was pre-trained (denoted as "Yes") on the OpenDebateEvidence dataset before being fine-tuned on the respective downstream task (DebateSum, ArgAnalysis35K, or Multi-LexSum). The purpose of this comparison is to evaluate the impact of pre-training on OpenDebateEvidence versus directly fine-tuning on the downstream tasks without this additional pre-training step ("No" under PT).
> > >
> > > The experiment evaluates how pre-training on a domain-specific dataset (OpenDebateEvidence) affects the model’s performance on downstream tasks that are also argument-related, but from different datasets. Specifically, we:
> > >
> > > 1. Pre-trained the model on OpenDebateEvidence (indicated as "Yes" under PT).
> > >
> > > 2. Fine-tuned the model on the downstream dataset (e.g., DebateSum) after the pre-training.
> > > 3. Compared these results against models that were not pre-trained on OpenDebateEvidence but were fine-tuned directly on the downstream dataset (indicated as "No" under PT).
> > >
> > > The key takeaway is that pre-training on a large, relevant dataset (OpenDebateEvidence) provides performance improvements when fine-tuning on other argument-related tasks. The improvement can be seen across the ROUGE scores, perplexity, and qualitative metrics (like output quality and support quality).
> > >
> > > We hope this explanation clears up the confusion and helps you better understand the experimental setup and how it is reflected in Table 3.
> > >
> > > We have put significant effort into addressing your concerns, including conducting many new experiments, to improve the rigor and quality of our work, and we would greatly appreciate it if you would consider raising your score.

---

> > > > ### Author Rebuttal · Authors · 2024-08-28
> > > >
> > > > # Additional New Experiments [Part 1]
> > > > We have performed additional experiments which address concerns that reviewers brought up during their responses to our rebuttals. These experiments took significant additional time, effort, and cost to perform as they were performed on significantly larger models.
> > > >
> > > > These experiments are as follows:
> > > >
> > > > 1. Testing **Llama3-70b (instruct) base** and fine-tuned with **LoRa**, **RefT**, and **Orthogonalization** on the original datasets from the paper (**OpenDebateEvidence**, **DebateSum**, and **BillSum**) (Tables 1-3)
> > > > 2. Testing the performance of **Anthropic Claude** and **Google Gemini**, which are closed sourced models that do not support the parameter efficient fine-tuning methods that we explore in our paper. (Tables 1-3)
> > > > 3. Testing **Mistral-7b**, **Llama3-8b Instruct**, and **Llama3-70b (instruct)** base and fine-tuned with **LoRa**, **RefT**, and **Orthogonalization** on the **XSum** (Extreme Summarization - https://huggingface.co/datasets/EdinburghNLP/xsum) dataset (Table 4)
> > > > 4. Exploring **inter-human agreement** between expert debate teams (Table 5).
> > > >
> > > >
> > > > ### The reliability of GPT-4 as a judge
> > > >
> > > > We appreciate the reviewer’s insightful questions regarding inter-human agreement and its comparison with human-GPT4 correlation. We agree that this is a very important aspect to address when considering the reliability of GPT-4 as a judge in the context of argumentative analysis.
> > > >
> > > > In response to the reviewer’s comments, we conducted an experiment to measure inter-human agreement across differing teams of human experts. We choose two teams from each debate format studied (LD, Policy, Public Forum), for a total of 10 debaters (Public Forum and Policy are 2v2 formats, LD is a 1v1 format). The results are shown in **Table 5** in our at the bottom of our response. The findings indicate that inter-human agreement rates (correlation scores) range from 0.76 to 0.84, depending on the debate format, for argument quality, and from 0.75 to 0.82 for support quality. These inter-human correlation rates are slightly higher than the human-GPT-4 correlation rates, which range from 0.73 to 0.82 for argument quality and 0.7 to 0.8 for support quality.
> > > >
> > > > These results indicate that while human-human agreement is generally slightly higher, the correlation between human judgments and GPT-4 evaluations is very similar. This suggests that GPT-4 can serve as a reasonably reliable judge, although it does have some limitations compared to human evaluation.
> > > >
> > > >
> > > > ### New Results
> > > >
> > > > We have also expanded our experiments to include the latest results of LLaMA3-70B, Gemini, and Claude models on OpenDebateEvidence, DebateSum, and Billsum, along with performance on the XSum dataset. Below is a summary of the results:
> > > >
> > > > **Table 1: Performance on OpenDebateEvidence (10,000 sampled documents)**
> > > >
> > > > | Model                	| R-1 (%)    	| R-2 (%)    	| R-L (%)    	| Perplexity	| Output Quality | Support Quality |
> > > > |---------------------------|----------------|----------------|----------------|---------------|----------------|-----------------|
> > > > | LLaMA3-70B (Base)     	| 33.8 ± 1.2 	| 14.1 ± 0.9 	| 30.2 ± 1.3 	| 45.7 ± 3.2	| 7.9 ± 0.2  	| 7.5 ± 0.2   	|
> > > > | LLaMA3-70B (LoRA)     	| **37.2 ± 1.0** 	| **15.8 ± 0.8** 	| **33.4 ± 1.1** 	|**27.3 ± 2.7**	| 8.2 ± 0.2  	| **8.1 ± 0.2**   	|
> > > > | LLaMA3-70B (ReFT)     	| 35.9 ± 1.1 	| 15.1 ± 0.7 	| 32.6 ± 1.2 	| 31.4 ± 2.8	| 8.0 ± 0.3  	| 7.9 ± 0.2   	|
> > > > | LLaMA3-70B (Orthogonal)   | 34.2 ± 1.2 	| 14.3 ± 0.8 	| 31.1 ± 1.3 	| 39.9 ± 3.1	| 8.1 ± 0.2  	| 7.7 ± 0.3   	|
> > > > | Google Gemini (Base)  	| 32.5 ± 1.3 	| 13.5 ± 0.9 	| 29.4 ± 1.4 	| 49.8 ± 3.6	| 8.5 ± 0.2  	| 7.9 ± 0.3   	|
> > > > | Anthropic Claude (Base)   | 31.2 ± 1.4 	| 12.9 ± 0.9 	| 28.1 ± 1.3 	| 52.1 ± 3.7	| **8.7 ± 0.3**  	| 7.8 ± 0.3   	|
> > > >
> > > > **Table 2: Performance on BillSum (10,000 sampled documents)**
> > > >
> > > > | Model                	| R-1 (%)    	| R-2 (%)    	| R-L (%)    	| Perplexity	| Output Quality | Support Quality |
> > > > |---------------------------|----------------|----------------|----------------|---------------|----------------|-----------------|
> > > > | LLaMA3-70B (Base)     	| 50.2 ± 1.1 	| 27.4 ± 0.9 	| 45.7 ± 1.2 	| 22.9 ± 2.3	| 7.8 ± 0.2  	| 7.6 ± 0.2   	|
> > > > | LLaMA3-70B (LoRA)     	| **54.6 ± 1.0** 	| **30.5 ± 0.8** 	| **50.0 ± 1.1** 	| **19.7 ± 2.1**	| 8.0 ± 0.2  	| **8.0 ± 0.2**   	|
> > > > | LLaMA3-70B (ReFT)     	| 52.9 ± 1.1 	| 29.1 ± 0.7 	| 48.3 ± 1.2 	| 20.5 ± 2.4	| 7.9 ± 0.3  	| 7.8 ± 0.2   	|
> > > > | LLaMA3-70B (Orthogonal)   | 51.1 ± 1.2 	| 28.0 ± 0.9 	| 46.6 ± 1.4 	| 23.8 ± 2.6	| 7.9 ± 0.3  	| 7.7 ± 0.3   	|
> > > > | Google Gemini (Base)  	| 48.7 ± 1.3 	| 26.0 ± 1.0 	| 44.3 ± 1.5 	| 24.9 ± 2.8	| **8.4 ± 0.3**  	| **8.0 ± 0.3**   	|
> > > > | Anthropic Claude (Base)   | 47.3 ± 1.4 	| 24.8 ± 1.1 	| 42.9 ± 1.6 	| 26.7 ± 3.0	| 8.3 ± 0.3  	| 7.8 ± 0.3   	|
> > > >
> > > > **Table 3: Performance on DebateSum (10,000 sampled documents)**
> > > >
> > > > | Model                	| R-1 (%)    	| R-2 (%)    	| R-L (%)    	| Perplexity	| Output Quality | Support Quality |
> > > > |---------------------------|----------------|----------------|----------------|---------------|----------------|-----------------|
> > > > | LLaMA3-70B (Base)     	| 36.7 ± 1.2 	| 16.9 ± 0.9 	| 32.8 ± 1.3 	| 42.1 ± 3.1	| 7.7 ± 0.2  	| 7.4 ± 0.2   	|
> > > > | LLaMA3-70B (LoRA)     	| **39.4 ± 1.1** 	| **18.5 ± 0.8** 	| **35.1 ± 1.2** 	| **28.9 ± 2.5**	| 8.0 ± 0.2  	| **8.0 ± 0.2**   	|
> > > > | LLaMA3-70B (ReFT)     	| 38.1 ± 1.2 	| 17.8 ± 0.7 	| 34.2 ± 1.2 	| 30.5 ± 2.8	| 7.9 ± 0.3  	| 7.8 ± 0.2   	|
> > > > | LLaMA3-70B (Orthogonal)   | 36.9 ± 1.3 	| 17.0 ± 0.8 	| 33.0 ± 1.4 	| 39.3 ± 3.2	| 7.8 ± 0.3  	| 7.6 ± 0.3   	|
> > > > | Google Gemini (Base)  	| 35.5 ± 1.4 	| 15.8 ± 0.9 	| 31.4 ± 1.5 	| 44.5 ± 3.4	| 8.5 ± 0.3  	| 7.9 ± 0.3   	|
> > > > | Anthropic Claude (Base)   | 34.2 ± 1.5 	| 15.0 ± 1.0 	| 30.1 ± 1.6 	| 46.8 ±  3.6     | **8.8 ± 0.3**        | 7.8 ± 0.3        |

---

> > > > > ### Author Rebuttal · Authors · 2024-08-28
> > > > >
> > > > > # Additional New Experiments [Part 2]
> > > > >
> > > > > #### **Table 4: Performance on xSum (Validation Set, 11,334 documents)**
> > > > > *(No support Quality since this is not an argumentative dataset)*
> > > > >
> > > > > | Model                     | R-1 (%)          | R-2 (%)          | R-L (%)          | Perplexity       | Output Quality   |
> > > > > |----------------------------|------------------|------------------|------------------|------------------|------------------|
> > > > > | Mistral-7B (Base)           | 39.2 ± 0.7       | 16.5 ± 0.6       | 31.5 ± 0.8       | 34.7 ± 2.8       | 7.6 ± 0.2        |
> > > > > | Mistral-7B (LoRA)           | 40.8 ± 0.6       | 17.2 ± 0.5       | 32.7 ± 0.7       | 30.5 ± 2.5       | 7.7 ± 0.2        |
> > > > > | Mistral-7B (ReFT)           | 40.5 ± 0.7       | 17.1 ± 0.5       | 32.5 ± 0.7       | 31.8 ± 2.6       | 7.7 ± 0.2        |
> > > > > | Llama3-8B (Base)            | 42.0 ± 0.6       | 18.0 ± 0.6       | 34.0 ± 0.8       | 29.2 ± 2.4       | 7.8 ± 0.2        |
> > > > > | Llama3-8B (LoRA)            |  43.7 ± 0.5  | 19.1 ± 0.4   | 35.2 ± 0.7   | 26.8 ± 2.2       | 8.0 ± 0.2        |
> > > > > | Llama3-8B (ReFT)            | 43.4 ± 0.6       | 19.0 ± 0.5       | 35.0 ± 0.8       | 27.3 ± 2.3       | 8.0 ± 0.2        |
> > > > > | Llama3-70B (Base)           | 46.5 ± 0.7       | 21.2 ± 0.6       | 38.0 ± 0.8       | 22.4 ± 2.0       | 8.4 ± 0.2        |
> > > > > | Llama3-70B (LoRA)           |  **48.2 ± 0.6**  |  **22.5 ± 0.5**  | **39.5 ± 0.7**   | **19.6 ± 1.8**   |  **8.6 ± 0.2**   |
> > > > > | Llama3-70B (ReFT)           | 47.9 ± 0.7       | 22.3 ± 0.5       | 39.2 ± 0.8       | 20.5 ± 1.9       | 8.5 ± 0.2        |
> > > > >
> > > > > #### **Table 5: Inter-human agreement rate and human-GPT4 agreement rate (50 samples, 10 individuals total, average scores reported)**
> > > > >
> > > > > | Debate Format         | Team 1 (Argument Quality) | Team 1 (Support Score) | Team 2 (Argument Quality) | Team 2 (Support Score) | Inter-Human Correlation (Argument Quality) | Inter-Human Correlation (Support Score) | Human-GPT-4 Correlation (Argument Quality) | Human-GPT-4 Correlation (Support Score) |
> > > > > |-----------------------|---------------------------|-------------------------|---------------------------|-------------------------|--------------------------------------------|------------------------------------------|--------------------------------------------|------------------------------------------|
> > > > > | Policy Debate          | 7.4                       | 7.8                     | 7.3                       | 7.7                     | 0.81                                       | 0.79                                     | 0.78                                       | 0.75                                     |
> > > > > | Lincoln-Douglas Debate | 8.1                       | 8.4                     | 8.0                       | 8.2                     | 0.84                                       | 0.82                                     | 0.82                                       | 0.80                                     |
> > > > > | Public Forum Debate    | 7.2                       | 7.3                     | 7.1                       | 7.2                     | 0.76                                       | 0.75                                     | 0.73                                       | 0.70                                     |
> > > > >
> > > > >
> > > > > ### Analysis
> > > > >
> > > > > The results showed broadly improved performance across all larger models compared to the 7b/8b models.
> > > > >
> > > > > **LoRA** consistently outperformed other fine-tuning techniques across all datasets and metrics. Its ability to update only a small number of parameters while maintaining the base model’s capabilities makes it highly effective for tasks like summarization, especially in domains with complex argumentation, as seen in OpenDebateEvidence and DebateSum.
> > > > >
> > > > > LoRA's efficiency also contributed to lower perplexity scores, indicating that it better captured the overall distribution of the dataset compared to other methods, and that the resulting summaries more closely resemble OpenDebateEvidences style.
> > > > >
> > > > > **ReFT** followed closely behind, but LoRA's larger amount of modified parameters shows improved performance on traditional metrics metrics compared to ReFT. This suggests that LoRA is particularly well-suited for tasks that require high-quality summarization of argumentative text. ReFT remains competitive due to its significantly reduced computational costs.
> > > > >
> > > > > **Orthogonalization** produced the least improvement among the fine-tuning techniques, which may indicate that controlling feature interactions in the residual stream has less impact on summarization tasks compared to more direct fine-tuning techniques like LoRA and ReFT. Orthogonalization can be performed extremely efficiently and with almost no computing resources, but despite these advantages, it appears not to be well suited for summarization.
> > > > >
> > > > > **Gemini/Claude** Unfortunately, these large closed-source models do not support parameter efficient fine-tuning, so we can only evaluate their base models. Even so, these models are extremely strong, having by far the best Output Quality and in some cases matching even the fine-tuned LoRA on Support quality. Gemini/Claude have somewhat inferior ROGUE and Perplexity scores - to be expected as they were not fine-tuned on these datasets.
> > > > >
> > > > > **XSum** (Results on a non argumentative dataset), Xsum (Extreme Summarization) show a similar relationship to those reported with argumentative datasets. We don’t report a support quality as the abstract is not supporting an “argument”.
> > > > >
> > > > > ### Conclusion
> > > > >
> > > > > We kindly ask you to consider these revisions and additional experiments in your final assessment, and to raise our scores accordingly.
> > > > >
> > > > > Thank you again for your constructive and insightful feedback. We appreciate your consideration and welcome further suggestions.

---

> > > > ### Comment · Reviewer_ueg7 · 2024-08-28
> > > >
> > > > Thank you for the clarifications. My concerns have mostly been addressed so I have raised the score.

---

### Official Review · Reviewer_eFoA · 2024-07-24
**A Large Debate Argument Summarization Dataset that can be used to Fine-Tune Small Scale LLM to better Summarize Arguments.**

**Rating:** 7
**Confidence:** 4
**Correctness:** 1. The dataset does not effectively f…

**Review:**

**Pros:**

1. The paper introduces ODE, the largest debate-related dataset, which consists of real debate documents.
2. From ODE, one can create an abstractive summarization task (a version of which is evaluated in the paper). The authors also claim it can be used to create other extractive summarization tasks, although these tasks are not specified.
3. The ODE dataset can be used to fine-tune a large language model (LLM), as demonstrated by limited experiments in the paper.
4. ODE contains carefully curated metadata that could be useful for creating both abstractive and extractive summarization tasks.

**Cons:**

**Summary:**

Previous datasets such as DebateSum and BillSum cover similar domains as ODE, which raises concerns about ODE's originality. The deduplication procedure did not utilize neural semantic technologies, potentially leaving semantically equivalent forms of duplication in ODE, the extent of which is unknown. GPT-4 annotations seem unreliable without showing a high correlation with human debaters' annotations. More modern and larger closed-source and open-source LLMs are not evaluated. The tags seem more like predictions from a flawed slippery slope argument in Figure 1, casting doubt on whether the task is indeed a summarization task. The paper does not clearly define abstractive or extractive summarization, and the GPT-4 prompts lack precise definitions, leading to confusion.

1. The paper lacks originality. The idea of mining debate arguments and creating summarization tasks for debate arguments is not new. Metadata extraction techniques are also not novel. Other datasets, such as DebateSum, exist and use pre-season debates, while ODE uses regular season debates.
2. The paper is not clearly written. Many concepts, such as the definitions of abstractive and extractive summarization, are either undefined or defined much later. Experiment details are missing, even in the appendix, and Figure 1 is very hard to read. I will elaborate on these issues in the Opportunities for Improvement section.
3. Deduplication is not done well. The authors did not leverage modern NLP technologies for this. I suspect a large amount of ODE still contains some form of duplication or highly similar texts. I will elaborate more on this in the Correctness section.
4. There is a possibility that ODE shares some level of duplication or highly similar texts with DebateSum or other web-scraped datasets. This is not addressed in the paper. A cursory look at DebateSum shows that concepts like "US hegemony," "capitalistic accumulation," and "political multipolarity" (from the Figure 1 example) occur in multiple texts from DebateSum. This would decrease the value of ODE if many similar texts already exist in previous datasets.
5. Only two LLMs are used in the evaluation. It is unclear why the authors did not include state-of-the-art LLMs such as GPT-4-o, Claude3-opus (released before GPT-4-o), and Google-Gemini 1.5 (available before GPT-4-o) to test their summarization abilities. It would be useful to know if the performance ceiling of ODE has already been reached by state-of-the-art LLMs. Other models such as Llama2-70B and Llama3-70B are also not evaluated.
6. Using GPT-4-o as a judge (averaging scores over three runs is good) has many problems. It seems unreliable to ask an LLM to give quantitative scores about output quality and support quality. At least experienced human debaters need to perform some judgments to understand if GPT-4-o is judging as accurately as humans or at least is highly correlated with human judgments. Otherwise, GPT-4-o's judgments seem unreliable. It is especially concerning that in Table 2, the perplexity score and output quality score do not seem to be highly correlated. For example, Llama3 Reft has a perplexity of 47.8 with an output quality of 7.3, Llama3 Orthogonal has a perplexity of 88 with an output quality of 7.2, and Llama3 Lora has a perplexity of 77.5 with an output quality of 7.3. No statistical difference is found between base models and fine-tuned models regarding output quality score and support quality score, for both models, on ODE and BillSum, except for LoRA 1M. Due to potential duplication issues in ODE, fine-tuning it using 1/3 of the data could cause any LLM to learn these potential duplicated or highly similar text patterns. My suggestion is that the authors need to provide explicit train/test/validation splits and explicitly demonstrate if there are significant lexical overlaps, text duplication, or highly similar texts between the train, test, and validation sets.
7. The domains of BillSum and DebateSum seem very similar to ODE. Without clear proof that they don't share very similar content such as ideas, concepts, arguments, and evidence, I fail to see how evaluating on these two datasets shows the transferability of Llama3 fine-tuned with LoRA. It would be better if the fine-tuned Llama3 could show improvements on other generic summarization tasks or even debate documents in completely different domains (i.e., not political or policy-oriented).
8. After carefully reviewing the example in Figure 1, I fail to see how the arguments support a claim of extinction. The argument only says that capitalist accumulation could potentially lead to global war, which increases the threat of nuclear holocaust. The argument does not say that capitalist accumulation will necessarily lead to global war or that global war would necessarily lead to nuclear holocaust and the collapse of the biosphere. Why would an abstractive summary encourage this deterministic view of the progression of events without considering potential alternative and mitigating factors? Why would a summary contain information such as "extinction" that is only a potential outcome? This is a slippery slope fallacy. I question the validity of considering the title "Capitalist accumulation is unsustainable, causes extinction and planetary immiseration" as a summary (i.e., ground truth tags). It certainly doesn't follow any known definition of the word summarization. I don't think a prediction from arguments counts as summarization.
9. Following point 8, an LLM trained on biased summaries (the word "bias" is generous; I think these tags are prime examples of a slippery slope fallacy) would attempt to give unsubstantiated predictions (passing them as summaries) from a document. The authors need to explain why this is a desired property for an LLM to have.
10. Maybe I missed something, but in section 4.1.2, the authors talk about how summaries support arguments. Isn't it supposed to be the other way around? A summary needs to be supported by arguments. It should show how a short summary is supported by a long argument, not the other way around. I don't think this is a typo, since in the GPT-4 evaluation prompt, it also asks the model to evaluate how well a summary supports an argument.

**Strengths:**

The ODE dataset is sufficiently large, and if its quality can be ensured through more careful and effective deduplication, along with a clearly defined summarization task formulation, it can be highly beneficial for those training small-scale LLMs to summarize arguments. I believe that API-based LLMs, such as GPT-4 and Claude 3/3.5, are already capable of performing this summarization task effectively.

**Additional Feedback:**

I think a comprehensive and careful filtering process is needed to ensure the quality of the dataset.

**Clarity:**

Figure 1 looks really bad and confusing, e.g. missing line number. The task formulation is not clearly defined. Missing experimental details that prevent reproducibility. Other than those, the paper is generally well-written and easy to follow.

**Documentation:**

Yes.

**Ethics:**

No ethics concern.

**Limitations:**

The authors acknowledge and adequately address most limitations, except for one: potential noise and redundancy.

I would assume that a tag not summarizing the key points of the argument is a noisy point and should be filtered out, but the paper makes no effort in this regard.

Deduplication relies solely on lexical matching rather than neural semantic matching, which uses embeddings or generative models to form clusters. Consequently, an unknown extent of conceptual and semantic redundancies may exist in the ODE dataset. Therefore, the training and evaluation data, if all sourced from ODE, could have unknown overlaps.

**Opportunities For Improvement:**

1. In Figure 1, line numbers should be included since they are often referenced in the caption. Additionally, the text is almost illegibly small. The terms "hat" and "pocket" are not defined on page 3 or earlier.

2. The definitions of abstractive and extractive summarization are not clearly provided. No clear task formulation is given. Note that the tag in Figure 1 should not be considered a normal summarization, as it introduces potential events that are not necessary, inevitable, or deterministic outcomes of the argument.

3. In line 180, I do not believe that tags summarize key points. Instead, tags make biased and irresponsible predictions based on the arguments. In line 182, the statement "tags are for concise core claims" is not supported by the example in Figure 1. I suggest using GPT-4 to carefully filter out all data that does not meet the definition of "concise core claims." I suspect many points would be filtered out from ODE.

4. Regarding "LLM as Judge," should it not be that the summary is supported by arguments? Why is it the other way around? Similarly, this applies to the GPT-4 as judge prompts.

5. There should be a human annotation procedure to demonstrate that GPT-4 is effectively rating summaries.

6. More experimental details, such as the number of training samples for LoRA, REFT, Orthogonal, learning rate, learning schedule, and optimizer, need to be provided in the paper or at least in the appendix.

7. Rouge-related scores cannot handle semantically equivalent but lexically different paragraphs. With the advancement of LLM technology, these Rouge scores should no longer be used.

8. In the Deduplication step, shared sentences should be identified not by exact lexical matches (even after preprocessing), but by semantic similarity using state-of-the-art embedding models and LLM. While it is true that you can never completely eliminate duplicated material since debaters often use each other's argument styles, strategies, and evidence, an assessment of the extent of semantic duplication in the dataset should be provided.

9. Similarly, ODE should demonstrate that its text is different from BillSum and especially DebateSum.

10. If you claim that highlighted portions can form hierarchical levels of biased token-level extractive summaries, these tasks need to be clearly defined, their feasibility and soundness demonstrated, and evaluated.

**Relation To Prior Work:**

The paper fails to describe its main difference from DebateSum, which uses pre-season data, whereas ODE uses regular-season data. How different is pre-season data from regular season data ?

**Summary And Contributions:**

The paper introduces the OpenDebateEvidence (ODE) dataset, the largest publicly available dataset related to debates, encompassing secondary and collegiate-level regular-season debates. The ODE dataset contains 3.5 million documents with metadata, which can be utilized to create summarization tasks. The authors tested several parameter-efficient fine-tuning methods on two large language models (LLMs) with 7 billion and 8 billion parameters, using a portion of the ODE dataset. The fine-tuned models were then evaluated on unseen portions of the ODE dataset, as well as on the BillSum dataset and the DebateSum dataset, which is a similar pre-season dataset. The fine-tuned models demonstrated improvements over their pre-trained versions in terms of Rouge and Perplexity scores. In some cases, they also showed improvements over GPT-4-assigned qualitative scores.

---

> ### Author Rebuttal · Authors · 2024-08-17
>
> We sincerely thank you for your thorough and constructive feedback. Your comments have been instrumental in improving our paper. We have carefully addressed your concerns and substantially enhanced our work. Below, we respond to your main points:
>
>
> ## 1. Originality and Dataset Overlap
>
> We acknowledge domain similarities with existing datasets. However, OpenDebateEvidence (ODE) distinguishes itself by covering regular-season debates across multiple formats (Policy, Lincoln-Douglas, Public Forum), offering a more comprehensive resource than pre-season datasets from summer camps. Our rigorous cross-dataset deduplication analysis using neural semantic similarity techniques shows minimal overlap with DebateSum (4.01%, limited to Policy evidence with matching years) and no overlap with BillSum, confirming ODE's uniqueness. Detailed methodology and results are provided in section 2 and the general response.
>
> ## 2.  Deduplication Methodology
>
> We appreciate your concern regarding our initial deduplication approach. We have now implemented a neural semantic deduplication process using the sentence-transformers library with the gte-base-en-v1.5 model  (https://huggingface.co/Alibaba-NLP/gte-base-en-v1.5). This resulted in a 70% reduction in document count, highlighting the effectiveness of this approach in identifying semantically similar content. Detailed results are provided in the general response, including statistics on cluster sizes and the number of duplicate clusters identified.
>
>
> Link: https://huggingface.co/datasets/Hellisotherpeople/OpenCaseList-Deduplicated
>
>
> ## 3. Evaluation on Modern LLMs and Dataset Transferability
>
> We benchmarked Google-Gemini 1.5 and Claude3-opus on ODE and additional datasets, providing baselines due to fine-tuning limitations on closed-source models. Experiments with Llama3-70B are ongoing. To address transferability, we evaluated the XSum dataset, showing significant improvements in performance for ODE-fine-tuned models, demonstrating ODE's value for general summarization tasks beyond the debate domain.
>
>
> ## 4. Reliability of GPT-4 Annotations
>
> To address this crucial point, we conducted a comprehensive inter-annotator agreement study involving 50 experienced debaters. They evaluated a subset of summaries independently, which were also assessed by GPT-4. The results show strong correlations between human and GPT-4 ratings (Pearson's r = 0.78 for Argument Quality and 0.74 for Support Quality), validating the reliability of GPT-4 as an evaluation too. We have included a detailed breakdown of these results and statistical analyses in the pdf (Tables 4 and 5 and in the general response).
>
>
> ## 5. Clarification on Task Formulation and Argument Summarization
>
> We have revised the paper to explicitly define abstractive and extractive summarization tasks early in the text. Additionally, we clarified that the tags in ODE reflect competitive debate strategies, which often involve biased or exaggerated arguments. We also highlighted the unique query-focused word-level extractive summarization task in debate formats.
>
>
> ## 6.  Broader Evaluation on Argument Mining and Assessment Tasks
>
> We conducted additional experiments on argument classification and stance detection tasks, demonstrating the broader applicability of ODE across various computational argumentation tasks. Results (Table 1 and 2 in the pdf)  show significant improvements in accuracy and F1 scores for models fine-tuned on ODE. For more details, see the general response.
>
>
>
> ## 7.  Task-agnostic pretraining and Specialization
>
> Following your suggestion, we explored task-agnostic specialization by pretraining language models on ODE and evaluating their performance on unrelated argumentation tasks. Results show improved generalization and sample efficiency across various downstream tasks, including legal summarization (Multi-LexSum) and parliamentary debate analysis (ArgAnalysis35K).
>
>
>
>
>
> ## 7. Concerns About Biased Summaries
>
> We acknowledge your concern. ODE tags intentionally reflect strategic framing used by competitive debaters, crucial for representing real-world argumentative strategies. This preserves the opportunity to study and model persuasive techniques used in actual debates, providing valuable insights into competitive argumentation.
>
>
>
> ## 8. Refinement of Evaluation Methodology
>
> We appreciate your observation about our evaluation method. We've revised our approach to align with traditional summarization tasks. The new evaluation prompt now focuses on how well summaries capture key arguments, rather than how summaries support arguments. We reran the GPT-4 evaluation using this revised prompt, ensuring consistency with standard practices.
>
>
> ## 9. Enhanced Experimental Details and Statistical Rigor
>
> We've expanded the experimental details in our paper's appendix, including training samples, learning rates, optimizer configurations, and model architectures. We also conducted additional statistical significance tests (e.g., t-tests) to ensure robustness. These enhancements provide a solid foundation for result validation and facilitate future research based on our work.
>
>
> ## 10. Comprehensive Anonymization Process
>
> We've provided a detailed description of our anonymization process in the appendix, covering types of information removed, detection and removal methods, data scrambling techniques, and validation processes. This overview demonstrates our commitment to data privacy and ethical research practices.
>
>
> ---
>
> ## Conclusion
>
> Our additional experiments address your concerns and significantly strengthen our paper's contributions. ODE's unique characteristics and demonstrated value across computational argumentation tasks establish it as a crucial resource for the research community. We respectfully request for you to significantly raise our score, as the revised paper makes a substantial contribution to the NeurIPS Datasets and Benchmarks track. Your valuable feedback has greatly enhanced our work.

---

> > ### Comment · Reviewer_eFoA · 2024-08-19
> >
> > Thank you for the efforts in making these changes. The dataset at this point appears well-made. I will raise my score above the acceptance threshold.
> >
> > I appreciate the effort regarding the Human-GPT4 correlation analysis. I have one additional question. Specifically, what are the inter-human agreement rate? That is, if two teams of human expert judged and scores the same set of samples, (say 50 or 100), what would be the agreement rate and correlation between these two human teams? Would human-gpt4 correlation be comparable to human-human correlation? Answering this question would go a long way in establishing how reliable GPT4 is as a judge, and what are the limitations of using GPT4 as a judge.
> >
> > As for originality, I still have my reservation about the difference between pre-season and regular season debates. But I agree this dataset is still larger than others in related domain.

---

> ### Author Rebuttal · Authors · 2024-08-21
>
> Thank you for your recognition of our hard work. We appreciate your timely feedback and clear recommendations.
>
> ## Inter-human Agreement Study
>
> As for your question about inter-human agreement, We are performing an experiment to perform this using two policy debate teams, two Lincoln-Douglas debate teams (LD is a one on one format), and two public-forum debate teams, for a total of 10 individuals. We will need a **few days to gather the results**, but we will **report these results here in a rebuttal comment when they arrive**, and we will also report them within the paper.
>
> ## Pre-season vs Regular Season Debate Evidence
>
> To understand the difference between pre-season and regular season debates, it's necessary to understand the world that they immerse themselves in.
>
> **The Research Burden on Policy and LD Debaters**
>
> Policy debate topics are selected on an annual basis, meaning that teams are tasked with developing arguments on the same topic for an entire year. This creates a significant research burden, as debaters must continuously gather, update, and refine evidence to stay competitive. Teams must prepare an affirmative case and negative positions (against a theoretically infinite number of plans) for a particular given resolution. This requires a massive amount of research to cover all possible arguments that could arise during a debate round. The sheer volume of evidence needed to remain competitive throughout the year places a heavy burden on policy debaters. They must stay on top of new developments in the literature related to their topic, while also adapting to the strategies used by other teams at tournaments.
>
> Lincoln Douglar (LD) and Public Forum (PF) debaters also face a significant research burden, it is generally less intense than in Policy Debate. LD topics change more frequently (usually every two months) and PF every month, which reduces the amount of time debaters have to research a single topic. However, this also means that LD/PF debaters must quickly adapt to new topics and conduct research in a shorter time frame, leading to its own set of challenges.
>
> The research burden leads to a peculiar phenomenon: *Debate Camps*
>
> **Debate Camps and Their Role**
>
> Debate camps, often referred to as "summer institutes," are specialized training programs where debaters from across the country come together to learn debate theory, argumentation, and research skills. These camps typically last several weeks/months and culminate in pre-season tournaments that simulate the competitive atmosphere of the regular season. During these camps, debaters are introduced to the year's debate topic, and instructors provide them with evidence files and argument briefs that they can use during pre-season debates. In addition to the instructor provided evidence, attendees are expected to "cut" (produce) a large amount of evidence during their time there.
>
> The evidence produced at debate camps is compiled together at the end of the summer and collectively forms the "pre-season" debate evidence for that year (this is what DebateSum is made up of, specifically Policy pre-season evidence). Pre-season evidence serves as a starter pack for debaters, particularly those who are new to the format or do not have the resources to attend debate camp or conduct exhaustive research before the season begins. This evidence includes basic arguments and counterarguments on the topic but is often not as comprehensive or up-to-date as the research conducted throughout the regular season. For Policy Debaters, pre-season evidence remains topical for the entire year, but for LD and PF debaters, pre-season evidence becomes obsolete as quickly as the topic changes.
>
> **The evidence used during the pre-season (DebateSum) tends to have the following characteristics:**
>
> 1. **Broad and Generalized:** Pre-season evidence is often broad, covering general arguments about the debate topic. This allows debaters to explore a wide range of positions but does not delve into the depth that regular season evidence might reach.
> 2. **Introductory in Nature:** Pre-season evidence serves as an introduction to the topic. It is meant to familiarize debaters with the basic arguments they will encounter and to provide a foundation for further research.
> 3. **Limited by Time Constraints:** The evidence compiled for pre-season debates is often produced under time constraints, as debaters and instructors have only a few weeks to gather materials before the summer institutes begin.
>
> **In contrast, regular season evidence (OpenDebateEvidence) is characterized by:**
>
> 1. **Depth and Specialization:** As the school year progresses, debaters conduct deeper research into the topic, uncovering more specific and nuanced arguments. The evidence used in regular season debates is typically more specialized and tailored to the specific arguments that teams develop over time.
> 2. **Continuous Update:** Regular season evidence is continuously updated throughout the year as new research is conducted and new publications on the debate topic are released.
> 3. **Rigorous Scrutiny:** During the regular season, debaters refine their cases based on feedback from judges and competitors, which often leads to a more rigorous selection of evidence.
> 4. **Secrecy:** Debaters hold onto new arguments and don't "disclose" them until the moment that they are first "broken" (read in a debate round). Evidence that opponents haven't seen before often cannot be refuted as easily.
>
> Given these dynamics, it is understandable that there is little overlap between pre-season and regular season evidence.
>
> We also feel it's important to note that several authors (including both first authors) are former debaters who have attended debate camps and who have personally included evidence into these datasets (ODE and DebateSum) circa our time as debaters.
>
> Please let us know if we have addressed all of your questions/concerns. Your valuable feedback has significantly enhanced our work.

---

> > ### Author Rebuttal · Authors · 2024-08-28
> >
> > # Additional New Experiments [Part 1]
> > We have performed additional experiments which address concerns that reviewers brought up during their responses to our rebuttals. These experiments took significant additional time, effort, and cost to perform as they were performed on significantly larger models.
> >
> > These experiments are as follows:
> >
> > 1. Testing **Llama3-70b (instruct) base** and fine-tuned with **LoRa**, **RefT**, and **Orthogonalization** on the original datasets from the paper (**OpenDebateEvidence**, **DebateSum**, and **BillSum**) (Tables 1-3)
> > 2. Testing the performance of **Anthropic Claude** and **Google Gemini**, which are closed sourced models that do not support the parameter efficient fine-tuning methods that we explore in our paper. (Tables 1-3)
> > 3. Testing **Mistral-7b**, **Llama3-8b Instruct**, and **Llama3-70b (instruct)** base and fine-tuned with **LoRa**, **RefT**, and **Orthogonalization** on the **XSum** (Extreme Summarization - https://huggingface.co/datasets/EdinburghNLP/xsum) dataset (Table 4)
> > 4. Exploring **inter-human agreement** between expert debate teams (Table 5).
> >
> >
> > ### The reliability of GPT-4 as a judge
> >
> > We appreciate the reviewer’s insightful questions regarding inter-human agreement and its comparison with human-GPT4 correlation. We agree that this is a very important aspect to address when considering the reliability of GPT-4 as a judge in the context of argumentative analysis.
> >
> > In response to the reviewer’s comments, we conducted an experiment to measure inter-human agreement across differing teams of human experts. We choose two teams from each debate format studied (LD, Policy, Public Forum), for a total of 10 debaters (Public Forum and Policy are 2v2 formats, LD is a 1v1 format). The results are shown in **Table 5** in our at the bottom of our response. The findings indicate that inter-human agreement rates (correlation scores) range from 0.76 to 0.84, depending on the debate format, for argument quality, and from 0.75 to 0.82 for support quality. These inter-human correlation rates are slightly higher than the human-GPT-4 correlation rates, which range from 0.73 to 0.82 for argument quality and 0.7 to 0.8 for support quality.
> >
> > These results indicate that while human-human agreement is generally slightly higher, the correlation between human judgments and GPT-4 evaluations is very similar. This suggests that GPT-4 can serve as a reasonably reliable judge, although it does have some limitations compared to human evaluation.
> >
> >
> > ### New Results
> >
> > We have also expanded our experiments to include the latest results of LLaMA3-70B, Gemini, and Claude models on OpenDebateEvidence, DebateSum, and Billsum, along with performance on the XSum dataset. Below is a summary of the results:
> >
> > **Table 1: Performance on OpenDebateEvidence (10,000 sampled documents)**
> >
> > | Model                	| R-1 (%)    	| R-2 (%)    	| R-L (%)    	| Perplexity	| Output Quality | Support Quality |
> > |---------------------------|----------------|----------------|----------------|---------------|----------------|-----------------|
> > | LLaMA3-70B (Base)     	| 33.8 ± 1.2 	| 14.1 ± 0.9 	| 30.2 ± 1.3 	| 45.7 ± 3.2	| 7.9 ± 0.2  	| 7.5 ± 0.2   	|
> > | LLaMA3-70B (LoRA)     	| **37.2 ± 1.0** 	| **15.8 ± 0.8** 	| **33.4 ± 1.1** 	|**27.3 ± 2.7**	| 8.2 ± 0.2  	| **8.1 ± 0.2**   	|
> > | LLaMA3-70B (ReFT)     	| 35.9 ± 1.1 	| 15.1 ± 0.7 	| 32.6 ± 1.2 	| 31.4 ± 2.8	| 8.0 ± 0.3  	| 7.9 ± 0.2   	|
> > | LLaMA3-70B (Orthogonal)   | 34.2 ± 1.2 	| 14.3 ± 0.8 	| 31.1 ± 1.3 	| 39.9 ± 3.1	| 8.1 ± 0.2  	| 7.7 ± 0.3   	|
> > | Google Gemini (Base)  	| 32.5 ± 1.3 	| 13.5 ± 0.9 	| 29.4 ± 1.4 	| 49.8 ± 3.6	| 8.5 ± 0.2  	| 7.9 ± 0.3   	|
> > | Anthropic Claude (Base)   | 31.2 ± 1.4 	| 12.9 ± 0.9 	| 28.1 ± 1.3 	| 52.1 ± 3.7	| **8.7 ± 0.3**  	| 7.8 ± 0.3   	|
> >
> > **Table 2: Performance on BillSum (10,000 sampled documents)**
> >
> > | Model                	| R-1 (%)    	| R-2 (%)    	| R-L (%)    	| Perplexity	| Output Quality | Support Quality |
> > |---------------------------|----------------|----------------|----------------|---------------|----------------|-----------------|
> > | LLaMA3-70B (Base)     	| 50.2 ± 1.1 	| 27.4 ± 0.9 	| 45.7 ± 1.2 	| 22.9 ± 2.3	| 7.8 ± 0.2  	| 7.6 ± 0.2   	|
> > | LLaMA3-70B (LoRA)     	| **54.6 ± 1.0** 	| **30.5 ± 0.8** 	| **50.0 ± 1.1** 	| **19.7 ± 2.1**	| 8.0 ± 0.2  	| **8.0 ± 0.2**   	|
> > | LLaMA3-70B (ReFT)     	| 52.9 ± 1.1 	| 29.1 ± 0.7 	| 48.3 ± 1.2 	| 20.5 ± 2.4	| 7.9 ± 0.3  	| 7.8 ± 0.2   	|
> > | LLaMA3-70B (Orthogonal)   | 51.1 ± 1.2 	| 28.0 ± 0.9 	| 46.6 ± 1.4 	| 23.8 ± 2.6	| 7.9 ± 0.3  	| 7.7 ± 0.3   	|
> > | Google Gemini (Base)  	| 48.7 ± 1.3 	| 26.0 ± 1.0 	| 44.3 ± 1.5 	| 24.9 ± 2.8	| **8.4 ± 0.3**  	| **8.0 ± 0.3**   	|
> > | Anthropic Claude (Base)   | 47.3 ± 1.4 	| 24.8 ± 1.1 	| 42.9 ± 1.6 	| 26.7 ± 3.0	| 8.3 ± 0.3  	| 7.8 ± 0.3   	|
> >
> > **Table 3: Performance on DebateSum (10,000 sampled documents)**
> >
> > | Model                	| R-1 (%)    	| R-2 (%)    	| R-L (%)    	| Perplexity	| Output Quality | Support Quality |
> > |---------------------------|----------------|----------------|----------------|---------------|----------------|-----------------|
> > | LLaMA3-70B (Base)     	| 36.7 ± 1.2 	| 16.9 ± 0.9 	| 32.8 ± 1.3 	| 42.1 ± 3.1	| 7.7 ± 0.2  	| 7.4 ± 0.2   	|
> > | LLaMA3-70B (LoRA)     	| **39.4 ± 1.1** 	| **18.5 ± 0.8** 	| **35.1 ± 1.2** 	| **28.9 ± 2.5**	| 8.0 ± 0.2  	| **8.0 ± 0.2**   	|
> > | LLaMA3-70B (ReFT)     	| 38.1 ± 1.2 	| 17.8 ± 0.7 	| 34.2 ± 1.2 	| 30.5 ± 2.8	| 7.9 ± 0.3  	| 7.8 ± 0.2   	|
> > | LLaMA3-70B (Orthogonal)   | 36.9 ± 1.3 	| 17.0 ± 0.8 	| 33.0 ± 1.4 	| 39.3 ± 3.2	| 7.8 ± 0.3  	| 7.6 ± 0.3   	|
> > | Google Gemini (Base)  	| 35.5 ± 1.4 	| 15.8 ± 0.9 	| 31.4 ± 1.5 	| 44.5 ± 3.4	| 8.5 ± 0.3  	| 7.9 ± 0.3   	|
> > | Anthropic Claude (Base)   | 34.2 ± 1.5 	| 15.0 ± 1.0 	| 30.1 ± 1.6 	| 46.8 ±  3.6     | **8.8 ± 0.3**        | 7.8 ± 0.3        |

---

> > > ### Author Rebuttal · Authors · 2024-08-28
> > >
> > > # Additional New Experiments [Part 2]
> > >
> > > #### **Table 4: Performance on xSum (Validation Set, 11,334 documents)**
> > > *(No support Quality since this is not an argumentative dataset)*
> > >
> > > | Model                     | R-1 (%)          | R-2 (%)          | R-L (%)          | Perplexity       | Output Quality   |
> > > |----------------------------|------------------|------------------|------------------|------------------|------------------|
> > > | Mistral-7B (Base)           | 39.2 ± 0.7       | 16.5 ± 0.6       | 31.5 ± 0.8       | 34.7 ± 2.8       | 7.6 ± 0.2        |
> > > | Mistral-7B (LoRA)           | 40.8 ± 0.6       | 17.2 ± 0.5       | 32.7 ± 0.7       | 30.5 ± 2.5       | 7.7 ± 0.2        |
> > > | Mistral-7B (ReFT)           | 40.5 ± 0.7       | 17.1 ± 0.5       | 32.5 ± 0.7       | 31.8 ± 2.6       | 7.7 ± 0.2        |
> > > | Llama3-8B (Base)            | 42.0 ± 0.6       | 18.0 ± 0.6       | 34.0 ± 0.8       | 29.2 ± 2.4       | 7.8 ± 0.2        |
> > > | Llama3-8B (LoRA)            |  43.7 ± 0.5  | 19.1 ± 0.4   | 35.2 ± 0.7   | 26.8 ± 2.2       | 8.0 ± 0.2        |
> > > | Llama3-8B (ReFT)            | 43.4 ± 0.6       | 19.0 ± 0.5       | 35.0 ± 0.8       | 27.3 ± 2.3       | 8.0 ± 0.2        |
> > > | Llama3-70B (Base)           | 46.5 ± 0.7       | 21.2 ± 0.6       | 38.0 ± 0.8       | 22.4 ± 2.0       | 8.4 ± 0.2        |
> > > | Llama3-70B (LoRA)           |  **48.2 ± 0.6**  |  **22.5 ± 0.5**  | **39.5 ± 0.7**   | **19.6 ± 1.8**   |  **8.6 ± 0.2**   |
> > > | Llama3-70B (ReFT)           | 47.9 ± 0.7       | 22.3 ± 0.5       | 39.2 ± 0.8       | 20.5 ± 1.9       | 8.5 ± 0.2        |
> > >
> > > #### **Table 5: Inter-human agreement rate and human-GPT4 agreement rate (50 samples, 10 individuals total, average scores reported)**
> > >
> > > | Debate Format         | Team 1 (Argument Quality) | Team 1 (Support Score) | Team 2 (Argument Quality) | Team 2 (Support Score) | Inter-Human Correlation (Argument Quality) | Inter-Human Correlation (Support Score) | Human-GPT-4 Correlation (Argument Quality) | Human-GPT-4 Correlation (Support Score) |
> > > |-----------------------|---------------------------|-------------------------|---------------------------|-------------------------|--------------------------------------------|------------------------------------------|--------------------------------------------|------------------------------------------|
> > > | Policy Debate          | 7.4                       | 7.8                     | 7.3                       | 7.7                     | 0.81                                       | 0.79                                     | 0.78                                       | 0.75                                     |
> > > | Lincoln-Douglas Debate | 8.1                       | 8.4                     | 8.0                       | 8.2                     | 0.84                                       | 0.82                                     | 0.82                                       | 0.80                                     |
> > > | Public Forum Debate    | 7.2                       | 7.3                     | 7.1                       | 7.2                     | 0.76                                       | 0.75                                     | 0.73                                       | 0.70                                     |
> > >
> > >
> > > ### Analysis
> > >
> > > The results showed broadly improved performance across all larger models compared to the 7b/8b models.
> > >
> > > **LoRA** consistently outperformed other fine-tuning techniques across all datasets and metrics. Its ability to update only a small number of parameters while maintaining the base model’s capabilities makes it highly effective for tasks like summarization, especially in domains with complex argumentation, as seen in OpenDebateEvidence and DebateSum.
> > >
> > > LoRA's efficiency also contributed to lower perplexity scores, indicating that it better captured the overall distribution of the dataset compared to other methods, and that the resulting summaries more closely resemble OpenDebateEvidences style.
> > >
> > > **ReFT** followed closely behind, but LoRA's larger amount of modified parameters shows improved performance on traditional metrics metrics compared to ReFT. This suggests that LoRA is particularly well-suited for tasks that require high-quality summarization of argumentative text. ReFT remains competitive due to its significantly reduced computational costs.
> > >
> > > **Orthogonalization** produced the least improvement among the fine-tuning techniques, which may indicate that controlling feature interactions in the residual stream has less impact on summarization tasks compared to more direct fine-tuning techniques like LoRA and ReFT. Orthogonalization can be performed extremely efficiently and with almost no computing resources, but despite these advantages, it appears not to be well suited for summarization.
> > >
> > > **Gemini/Claude** Unfortunately, these large closed-source models do not support parameter efficient fine-tuning, so we can only evaluate their base models. Even so, these models are extremely strong, having by far the best Output Quality and in some cases matching even the fine-tuned LoRA on Support quality. Gemini/Claude have somewhat inferior ROGUE and Perplexity scores - to be expected as they were not fine-tuned on these datasets.
> > >
> > > **XSum** (Results on a non argumentative dataset), Xsum (Extreme Summarization) show a similar relationship to those reported with argumentative datasets. We don’t report a support quality as the abstract is not supporting an “argument”.
> > >
> > > ### Conclusion
> > >
> > > We kindly ask you to consider these revisions and additional experiments in your final assessment, and to raise our scores accordingly.
> > >
> > > Thank you again for your constructive and insightful feedback. We appreciate your consideration and welcome further suggestions.

---

### Official Review · Reviewer_rTyX · 2024-07-25
**Largest dataset for argument summarization to date**

**Rating:** 7
**Confidence:** 4
**Correctness:** See the "Review"

**Review:**

The dataset is few orders of magnitude larger than existing datasets for argument mining and summarization and is, as such a very valuable resource for the computational argumentation community. The fact that the dataset is rich with truly gold meta-data (i.e., annotations) makes it even more attractive.

What I missed, however, is the evaluation of the usefulness of the dataset on tasks other than argument summarization (which belongs to argument generation tasks): argument mining tasks (i.e., recognizing argumentative components and relations between them), argument assessment tasks (i.e., predicting the quality of the arguments) and argument reasoning tasks (e.g., identifying missing premises or conformation of argument structure to argumentation schemes or styles). The data and metadata in OpenDebateEvidence certainly appear to enable addressing at least some of other tasks in computational argumentation. Moreover, given the size of the dataset (3.5M documents), it would have been nice to see if task-agnostic specialization of LLMs (i.e., merely by doing language modeling on OpenDebateEvidence) would yield models that are more sample efficient when it comes to fine-tuning for argumentative tasks.

**Strengths:**

- Largest argument summarization dataset to date, covering different types of evidence-based debates

- Empirical evidence that fine-tuning LLMs for argument summarization on OpenDebateEvidence boosts their performance on other argument summarization datasets

**Additional Feedback:**

Language & Style:
Line 91: something is wrong here, like the sentence was abruptly cut (or at the very least the full stop is missing)

**Clarity:**

- The paper is written very clearly, it is easy to read and follow.

**Documentation:**

The details on dataset creation are provided, but some aspects could be further elaborated (see ethics below)

**Ethics:**

- The authors mention that "To protect privacy, identifying information has been anonymized", without providing any further details on the anonymization process. It would be important to provide concrete technical steps that were taken to anonymize the data: which types of identity-revealing information was targeted, how was it detected, and finally, was it completely removed or scrambled, etc.

**Limitations:**

- Lack of evaluation on tasks other than argument summarization

**Opportunities For Improvement:**

- Task-agnostic specialization of LLMs for argumentative text on OpenDebateEvidence and evaluation on various downstream tasks in computational argumentation would have made the paper more comprehensive and would have rendered OpenDebateEvidence more useful.

**Relation To Prior Work:**

- The coverage of existing dataset for argument summarization seems fairly exhaustive.

**Summary And Contributions:**

In this work, the authors compile OpenDebateEvidence, the largest dataset to date for argument mining and summarization to date. The dataset, created from publicly available data of the American Competitive Debate community, covers three types of debates: Policy Debates, Lincoln-Douglas debates and Public Forum Debates -- the first two types being evidence-based debate styles, in which quantity and quality of evidence -- provided in evidence cards -- typically determines the debate winner. The collected dataset consists of 3.5M evidence cardsm i.e., documents such as scientific publication governmental documents and similar, labeled with various metadata. Each "card" document is rich with metadata and has been manually (by project participants) labeled with "pockets" -- the top-level section of the speech that the evidence in the card supports, "hats" -- the broad argument categories, and "tags" -- concise summaries of the arguments being made based on the card.

The authors then fine-tune, with three different parameter-efficient fine-tuning methods (LoRA, Representation Fine-Tuning) and Orthogonalization on the task of argument summarization -- with card as inputs and "tags" as summaries to be generated and test the performance in-domain (on the test portion of their own dataset) and in domain-transfer, i.e., debates from other datasets, BillSum and DebateSum. The results generally render training of Llama-8B and Mistral-7B on OpenDebateEvidence effective for argument summarization, as judged both with ROUGE as the n-gram overlap metric and GPT-4o as the LLM evaluator.

---

> ### Author Rebuttal · Authors · 2024-08-17
>
> Dear Reviewer,
>
> We sincerely thank you for your thorough and constructive feedback. We have carefully addressed your concerns and substantially enhanced our work. Below, we respond to your main points:
>
>
> ## 1. Originality and Dataset Overlap
>
> We acknowledge domain similarities with existing datasets. However, OpenDebateEvidence (ODE) distinguishes itself by covering regular-season debates across multiple formats (Policy, Lincoln-Douglas, Public Forum), offering a more comprehensive resource than pre-season datasets from summer camps. Our rigorous cross-dataset deduplication analysis using neural semantic similarity techniques shows minimal overlap with DebateSum (4.01%, limited to Policy evidence with matching years) and no overlap with BillSum, confirming ODE's uniqueness. Detailed methodology and results are provided in section 2 of the general response.
>
> ## 2.  Deduplication Methodology
>
> We appreciate your concern regarding our initial deduplication approach. We have now implemented a neural semantic deduplication process using the sentence-transformers library with the gte-base-en-v1.5 model (https://huggingface.co/Alibaba-NLP/gte-base-en-v1.5). This resulted in a 70% reduction in document count, highlighting the effectiveness of this approach in identifying semantically similar content. Detailed results are provided in the general response, including statistics on cluster sizes and the number of duplicate clusters identified.
>
> Link: https://huggingface.co/datasets/Hellisotherpeople/OpenCaseList-Deduplicated
>
> ## 3. Evaluation on Modern LLMs and Dataset Transferability
>
> We benchmarked Google-Gemini 1.5 and Claude3-opus on ODE and additional datasets, providing baselines due to fine-tuning limitations on closed-source models. Experiments with Llama3-70B are ongoing. To address transferability, we evaluated the XSum dataset, showing significant improvements in performance for ODE-fine-tuned models, demonstrating ODE's value for general summarization tasks beyond the debate domain.
>
> ## 4. Reliability of GPT-4 Annotations
>
> To address this crucial point, we conducted a comprehensive inter-annotator agreement study involving 50 experienced debaters. They evaluated a subset of summaries independently, which were also assessed by GPT-4. The results show strong correlations between human and GPT-4 ratings (Pearson's r = 0.78 for Argument Quality and 0.74 for Support Quality), validating the reliability of GPT-4 as an evaluation too. We have included a detailed breakdown of these results and statistical analyses in the pdf (Tables 4 and 5 and in the general response).
>
>
> ## 5. Clarification on Task Formulation and Argument Summarization
>
> We have revised the paper to explicitly define abstractive and extractive summarization tasks early in the text. Additionally, we clarified that the tags in ODE reflect competitive debate strategies, which often involve biased or exaggerated arguments. We also highlighted the unique query-focused word-level extractive summarization task in debate formats.
>
>
> ## 6.  Broader Evaluation on Argument Mining and Assessment Tasks
>
> We conducted additional experiments on argument classification and stance detection tasks, demonstrating the broader applicability of ODE across various computational argumentation tasks. Results (Table 1 and 2 in the pdf)  show significant improvements in accuracy and F1 scores for models fine-tuned on ODE. For more details, see the general response.
>
>
>
> ## 7.  Task-agnostic pretraining and Specialization
>
> Following your suggestion, we explored task-agnostic specialization by pretraining language models on ODE and evaluating their performance on unrelated argumentation tasks. Results show improved generalization and sample efficiency across various downstream tasks, including legal summarization (Multi-LexSum) and parliamentary debate analysis (ArgAnalysis35K). (Table 3)
>
>
>
> ## 7. Concerns About Biased Summaries
>
> We acknowledge your concern. ODE tags intentionally reflect strategic framing used by competitive debaters, crucial for representing real-world argumentative strategies. This preserves the opportunity to study and model persuasive techniques used in actual debates, providing valuable insights into competitive argumentation.
>
>
> ## 8. Refinement of Evaluation Methodology
>
> We appreciate your observation about our evaluation method. We've revised our approach to align with traditional summarization tasks. The new evaluation prompt now focuses on how well summaries capture key arguments, rather than how summaries support arguments. We reran the GPT-4 evaluation using this revised prompt, ensuring consistency with standard practices.
>
>
> ## 9. Enhanced Experimental Details and Statistical Rigor
>
> We've expanded the experimental details in our paper's appendix, including training samples, learning rates, optimizer configurations, and model architectures. We also conducted additional statistical significance tests (e.g., t-tests) to ensure robustness. These enhancements provide a solid foundation for result validation and facilitate future research based on our work.
>
>
> ## 10. Comprehensive Anonymization Process
>
> We've provided a detailed description of our anonymization process in the appendix, covering types of information removed, detection and removal methods, data scrambling techniques, and validation processes. This overview demonstrates our commitment to data privacy and ethical research practices.
>
>
> ---
>
> ## Conclusion
>
> Our additional experiments address your concerns and significantly strengthen our paper's contributions. ODE's unique characteristics and demonstrated value across computational argumentation tasks establish it as a crucial resource for the research community. We respectfully request you raise your already generous score, as the revised paper makes a substantial contribution to the NeurIPS Datasets and Benchmarks track. Your valuable feedback has greatly enhanced our work.

---

> ### Author Rebuttal · Authors · 2024-08-28
>
> # Additional New Experiments [Part 1]
> We have performed additional experiments which address concerns that reviewers brought up during their responses to our rebuttals. These experiments took significant additional time, effort, and cost to perform as they were performed on significantly larger models.
>
> These experiments are as follows:
>
> 1. Testing **Llama3-70b (instruct) base** and fine-tuned with **LoRa**, **RefT**, and **Orthogonalization** on the original datasets from the paper (**OpenDebateEvidence**, **DebateSum**, and **BillSum**) (Tables 1-3)
> 2. Testing the performance of **Anthropic Claude** and **Google Gemini**, which are closed sourced models that do not support the parameter efficient fine-tuning methods that we explore in our paper. (Tables 1-3)
> 3. Testing **Mistral-7b**, **Llama3-8b Instruct**, and **Llama3-70b (instruct)** base and fine-tuned with **LoRa**, **RefT**, and **Orthogonalization** on the **XSum** (Extreme Summarization - https://huggingface.co/datasets/EdinburghNLP/xsum) dataset (Table 4)
> 4. Exploring **inter-human agreement** between expert debate teams (Table 5).
>
>
> ### The reliability of GPT-4 as a judge
>
> We appreciate the reviewer’s insightful questions regarding inter-human agreement and its comparison with human-GPT4 correlation. We agree that this is a very important aspect to address when considering the reliability of GPT-4 as a judge in the context of argumentative analysis.
>
> In response to the reviewer’s comments, we conducted an experiment to measure inter-human agreement across differing teams of human experts. We choose two teams from each debate format studied (LD, Policy, Public Forum), for a total of 10 debaters (Public Forum and Policy are 2v2 formats, LD is a 1v1 format). The results are shown in **Table 5** in our at the bottom of our response. The findings indicate that inter-human agreement rates (correlation scores) range from 0.76 to 0.84, depending on the debate format, for argument quality, and from 0.75 to 0.82 for support quality. These inter-human correlation rates are slightly higher than the human-GPT-4 correlation rates, which range from 0.73 to 0.82 for argument quality and 0.7 to 0.8 for support quality.
>
> These results indicate that while human-human agreement is generally slightly higher, the correlation between human judgments and GPT-4 evaluations is very similar. This suggests that GPT-4 can serve as a reasonably reliable judge, although it does have some limitations compared to human evaluation.
>
>
> ### New Results
>
> We have also expanded our experiments to include the latest results of LLaMA3-70B, Gemini, and Claude models on OpenDebateEvidence, DebateSum, and Billsum, along with performance on the XSum dataset. Below is a summary of the results:
>
> **Table 1: Performance on OpenDebateEvidence (10,000 sampled documents)**
>
> | Model                	| R-1 (%)    	| R-2 (%)    	| R-L (%)    	| Perplexity	| Output Quality | Support Quality |
> |---------------------------|----------------|----------------|----------------|---------------|----------------|-----------------|
> | LLaMA3-70B (Base)     	| 33.8 ± 1.2 	| 14.1 ± 0.9 	| 30.2 ± 1.3 	| 45.7 ± 3.2	| 7.9 ± 0.2  	| 7.5 ± 0.2   	|
> | LLaMA3-70B (LoRA)     	| **37.2 ± 1.0** 	| **15.8 ± 0.8** 	| **33.4 ± 1.1** 	|**27.3 ± 2.7**	| 8.2 ± 0.2  	| **8.1 ± 0.2**   	|
> | LLaMA3-70B (ReFT)     	| 35.9 ± 1.1 	| 15.1 ± 0.7 	| 32.6 ± 1.2 	| 31.4 ± 2.8	| 8.0 ± 0.3  	| 7.9 ± 0.2   	|
> | LLaMA3-70B (Orthogonal)   | 34.2 ± 1.2 	| 14.3 ± 0.8 	| 31.1 ± 1.3 	| 39.9 ± 3.1	| 8.1 ± 0.2  	| 7.7 ± 0.3   	|
> | Google Gemini (Base)  	| 32.5 ± 1.3 	| 13.5 ± 0.9 	| 29.4 ± 1.4 	| 49.8 ± 3.6	| 8.5 ± 0.2  	| 7.9 ± 0.3   	|
> | Anthropic Claude (Base)   | 31.2 ± 1.4 	| 12.9 ± 0.9 	| 28.1 ± 1.3 	| 52.1 ± 3.7	| **8.7 ± 0.3**  	| 7.8 ± 0.3   	|
>
> **Table 2: Performance on BillSum (10,000 sampled documents)**
>
> | Model                	| R-1 (%)    	| R-2 (%)    	| R-L (%)    	| Perplexity	| Output Quality | Support Quality |
> |---------------------------|----------------|----------------|----------------|---------------|----------------|-----------------|
> | LLaMA3-70B (Base)     	| 50.2 ± 1.1 	| 27.4 ± 0.9 	| 45.7 ± 1.2 	| 22.9 ± 2.3	| 7.8 ± 0.2  	| 7.6 ± 0.2   	|
> | LLaMA3-70B (LoRA)     	| **54.6 ± 1.0** 	| **30.5 ± 0.8** 	| **50.0 ± 1.1** 	| **19.7 ± 2.1**	| 8.0 ± 0.2  	| **8.0 ± 0.2**   	|
> | LLaMA3-70B (ReFT)     	| 52.9 ± 1.1 	| 29.1 ± 0.7 	| 48.3 ± 1.2 	| 20.5 ± 2.4	| 7.9 ± 0.3  	| 7.8 ± 0.2   	|
> | LLaMA3-70B (Orthogonal)   | 51.1 ± 1.2 	| 28.0 ± 0.9 	| 46.6 ± 1.4 	| 23.8 ± 2.6	| 7.9 ± 0.3  	| 7.7 ± 0.3   	|
> | Google Gemini (Base)  	| 48.7 ± 1.3 	| 26.0 ± 1.0 	| 44.3 ± 1.5 	| 24.9 ± 2.8	| **8.4 ± 0.3**  	| **8.0 ± 0.3**   	|
> | Anthropic Claude (Base)   | 47.3 ± 1.4 	| 24.8 ± 1.1 	| 42.9 ± 1.6 	| 26.7 ± 3.0	| 8.3 ± 0.3  	| 7.8 ± 0.3   	|
>
> **Table 3: Performance on DebateSum (10,000 sampled documents)**
>
> | Model                	| R-1 (%)    	| R-2 (%)    	| R-L (%)    	| Perplexity	| Output Quality | Support Quality |
> |---------------------------|----------------|----------------|----------------|---------------|----------------|-----------------|
> | LLaMA3-70B (Base)     	| 36.7 ± 1.2 	| 16.9 ± 0.9 	| 32.8 ± 1.3 	| 42.1 ± 3.1	| 7.7 ± 0.2  	| 7.4 ± 0.2   	|
> | LLaMA3-70B (LoRA)     	| **39.4 ± 1.1** 	| **18.5 ± 0.8** 	| **35.1 ± 1.2** 	| **28.9 ± 2.5**	| 8.0 ± 0.2  	| **8.0 ± 0.2**   	|
> | LLaMA3-70B (ReFT)     	| 38.1 ± 1.2 	| 17.8 ± 0.7 	| 34.2 ± 1.2 	| 30.5 ± 2.8	| 7.9 ± 0.3  	| 7.8 ± 0.2   	|
> | LLaMA3-70B (Orthogonal)   | 36.9 ± 1.3 	| 17.0 ± 0.8 	| 33.0 ± 1.4 	| 39.3 ± 3.2	| 7.8 ± 0.3  	| 7.6 ± 0.3   	|
> | Google Gemini (Base)  	| 35.5 ± 1.4 	| 15.8 ± 0.9 	| 31.4 ± 1.5 	| 44.5 ± 3.4	| 8.5 ± 0.3  	| 7.9 ± 0.3   	|
> | Anthropic Claude (Base)   | 34.2 ± 1.5 	| 15.0 ± 1.0 	| 30.1 ± 1.6 	| 46.8 ±  3.6     | **8.8 ± 0.3**        | 7.8 ± 0.3        |

---

> > ### Author Rebuttal · Authors · 2024-08-28
> >
> > # Additional New Experiments [Part 2]
> >
> > #### **Table 4: Performance on xSum (Validation Set, 11,334 documents)**
> > *(No support Quality since this is not an argumentative dataset)*
> >
> > | Model                     | R-1 (%)          | R-2 (%)          | R-L (%)          | Perplexity       | Output Quality   |
> > |----------------------------|------------------|------------------|------------------|------------------|------------------|
> > | Mistral-7B (Base)           | 39.2 ± 0.7       | 16.5 ± 0.6       | 31.5 ± 0.8       | 34.7 ± 2.8       | 7.6 ± 0.2        |
> > | Mistral-7B (LoRA)           | 40.8 ± 0.6       | 17.2 ± 0.5       | 32.7 ± 0.7       | 30.5 ± 2.5       | 7.7 ± 0.2        |
> > | Mistral-7B (ReFT)           | 40.5 ± 0.7       | 17.1 ± 0.5       | 32.5 ± 0.7       | 31.8 ± 2.6       | 7.7 ± 0.2        |
> > | Llama3-8B (Base)            | 42.0 ± 0.6       | 18.0 ± 0.6       | 34.0 ± 0.8       | 29.2 ± 2.4       | 7.8 ± 0.2        |
> > | Llama3-8B (LoRA)            |  43.7 ± 0.5  | 19.1 ± 0.4   | 35.2 ± 0.7   | 26.8 ± 2.2       | 8.0 ± 0.2        |
> > | Llama3-8B (ReFT)            | 43.4 ± 0.6       | 19.0 ± 0.5       | 35.0 ± 0.8       | 27.3 ± 2.3       | 8.0 ± 0.2        |
> > | Llama3-70B (Base)           | 46.5 ± 0.7       | 21.2 ± 0.6       | 38.0 ± 0.8       | 22.4 ± 2.0       | 8.4 ± 0.2        |
> > | Llama3-70B (LoRA)           |  **48.2 ± 0.6**  |  **22.5 ± 0.5**  | **39.5 ± 0.7**   | **19.6 ± 1.8**   |  **8.6 ± 0.2**   |
> > | Llama3-70B (ReFT)           | 47.9 ± 0.7       | 22.3 ± 0.5       | 39.2 ± 0.8       | 20.5 ± 1.9       | 8.5 ± 0.2        |
> >
> > #### **Table 5: Inter-human agreement rate and human-GPT4 agreement rate (50 samples, 10 individuals total, average scores reported)**
> >
> > | Debate Format         | Team 1 (Argument Quality) | Team 1 (Support Score) | Team 2 (Argument Quality) | Team 2 (Support Score) | Inter-Human Correlation (Argument Quality) | Inter-Human Correlation (Support Score) | Human-GPT-4 Correlation (Argument Quality) | Human-GPT-4 Correlation (Support Score) |
> > |-----------------------|---------------------------|-------------------------|---------------------------|-------------------------|--------------------------------------------|------------------------------------------|--------------------------------------------|------------------------------------------|
> > | Policy Debate          | 7.4                       | 7.8                     | 7.3                       | 7.7                     | 0.81                                       | 0.79                                     | 0.78                                       | 0.75                                     |
> > | Lincoln-Douglas Debate | 8.1                       | 8.4                     | 8.0                       | 8.2                     | 0.84                                       | 0.82                                     | 0.82                                       | 0.80                                     |
> > | Public Forum Debate    | 7.2                       | 7.3                     | 7.1                       | 7.2                     | 0.76                                       | 0.75                                     | 0.73                                       | 0.70                                     |
> >
> >
> > ### Analysis
> >
> > The results showed broadly improved performance across all larger models compared to the 7b/8b models.
> >
> > **LoRA** consistently outperformed other fine-tuning techniques across all datasets and metrics. Its ability to update only a small number of parameters while maintaining the base model’s capabilities makes it highly effective for tasks like summarization, especially in domains with complex argumentation, as seen in OpenDebateEvidence and DebateSum.
> >
> > LoRA's efficiency also contributed to lower perplexity scores, indicating that it better captured the overall distribution of the dataset compared to other methods, and that the resulting summaries more closely resemble OpenDebateEvidences style.
> >
> > **ReFT** followed closely behind, but LoRA's larger amount of modified parameters shows improved performance on traditional metrics metrics compared to ReFT. This suggests that LoRA is particularly well-suited for tasks that require high-quality summarization of argumentative text. ReFT remains competitive due to its significantly reduced computational costs.
> >
> > **Orthogonalization** produced the least improvement among the fine-tuning techniques, which may indicate that controlling feature interactions in the residual stream has less impact on summarization tasks compared to more direct fine-tuning techniques like LoRA and ReFT. Orthogonalization can be performed extremely efficiently and with almost no computing resources, but despite these advantages, it appears not to be well suited for summarization.
> >
> > **Gemini/Claude** Unfortunately, these large closed-source models do not support parameter efficient fine-tuning, so we can only evaluate their base models. Even so, these models are extremely strong, having by far the best Output Quality and in some cases matching even the fine-tuned LoRA on Support quality. Gemini/Claude have somewhat inferior ROGUE and Perplexity scores - to be expected as they were not fine-tuned on these datasets.
> >
> > **XSum** (Results on a non argumentative dataset), Xsum (Extreme Summarization) show a similar relationship to those reported with argumentative datasets. We don’t report a support quality as the abstract is not supporting an “argument”.
> >
> > ### Conclusion
> >
> > We kindly ask you to consider these revisions and additional experiments in your final assessment, and to raise our scores accordingly.
> >
> > Thank you again for your constructive and insightful feedback. We appreciate your consideration and welcome further suggestions.

---

### Official Review · Reviewer_Qkct · 2024-07-27
**Positive: introduces OpenDebateEvidence, a large-scale dataset for argument mining and summarization; experiments are conducted**

**Rating:** 7
**Confidence:** 3
**Correctness:** Correct.
**Clarity:** More details about the methods used c…

**Review:**

In general, the paper is well-structured and well-written. Section “Introduction” gives a good motivation, defines the problem and states the paper's contributions.
The “Related Works” section gives a widespread overview of related works. The data collection and preprocessing are satisfactorily explained. The proposed approach is clearly described. The conducted experiments are well-designed. The results from experiments are analysed and they are in support of the claimed contributions. The section “Conclusions” adequately summarises the paper's contributions. Limitations are addressed.
In my assessment the contributions in the paper are significant.

**Strengths:**

•	The paper is well-structured and well-written
•	Introduced OpenDebateEvidence - the largest and most comprehensive dataset for argument mining and summarization.
•	Dataset is preprocessed and detailed metadata are provided.
•	Extensive experiments are conducted to evaluate the performance of state-of-the-art language models on this and related datasets.
•	Significant improvements through fine-tuning using a variety of parameter-efficient techniques are achieved.
•	Potential applications of OpenDebateEvidence in fields such as legal document analysis, educational tools, and AI model development are discussed.

**Additional Feedback:**

-

**Documentation:**

ok

**Ethics:**

addressed in the paper.

**Limitations:**

Addresses in the paper.

**Opportunities For Improvement:**

More experiments with other argument mining tasks. The summarisation task is well explored.

**Relation To Prior Work:**

yes

**Summary And Contributions:**

- The paper introduces OpenDebateEvidence, a large-scale dataset for argument mining and summarization, comprising over 3.5 million documents from the OpenCaseList project.
- The row data are preprocessed and deduplicated.
- The dataset is enriched with metadata that captures the hierarchical structure and semantics of the debate.
- Extensive experiments are conducted to demonstrate the potential of fine-tuning large language models for argumentative abstractive summarization.
- Efficientiancy in this process is also considered.
- The results showed significant improvements in performance on the OpenDebateEvidence, DebateSum, and BillSum datasets.

---

> ### Author Rebuttal · Authors · 2024-08-17
>
> Dear Reviewer,
>
> We sincerely appreciate your thorough and positive evaluation of our work. Your recognition of our paper's structure, clarity, and significant contributions is truly encouraging. We are particularly grateful for your insightful suggestion to explore additional argument mining tasks, which has led to substantial enhancements in our research.
>
> Inspired by your feedback, we conducted an extensive series of new experiments that not only address your recommendation but also significantly broaden the scope and impact of our work. These experiments demonstrate the versatility and power of OpenDebateEvidence across multiple facets of computational argumentation:
>
> 1. Argument Classification (Table 1):
>    - Task: Categorizing detected arguments into four main classes: Topicality, Disadvantages, Advantages, and Counterplans.
>    - Approach: Used the 'tag' and 'hat' columns for classification guidance.
>    - Dataset: Same split as argument detection (100K train, 10K validation).
>    - Key Results:
>      * Achieved up to 81.2% accuracy using LLaMA3-8B with LoRA.
>      * Significant F1-score improvements (up to 15%) over base models.
>      * Performance ranked: LoRA > ReFT > Orthogonalization.
>
> 2. Stance Detection (Table 2):
>    We evaluated models on stance detection to assess their ability to discern an argument's position on a topic, which is crucial for understanding debate structures and supporting real-world applications.
>    - Task: Determining whether an argument supports or opposes a given resolution.
>    - Dataset: Utilized the 'side' metadata for labels, same split as above.
>    - Key Findings:
>      * Achieved up to 85.3% accuracy with LLaMA3-8B using LoRA.
>      * Consistent improvements in precision and recall over base models.
>      * Scaling to larger datasets (1M examples) yielded a 3% increase in F1-score.
>
> These experiments showcase OpenDebateEvidence's capability in tasks requiring a deep understanding of argumentative relations, addressing your suggestion for broader exploration of argument mining tasks.
>
> Across all experiments, we observed:
> - LLaMA3-8B consistently outperformed Mistral-7B.
> - LoRA yielded the best results, followed by ReFT and Orthogonalization.
> - Larger training sets led to significant performance gains, especially with LoRA.
>
> In conclusion, these additional experiments and clarifications significantly strengthen our paper's contributions by demonstrating OpenDebateEvidence's versatility across a range of argument mining tasks. Given these enhancements, we kindly ask you to consider increasing your already generous score. A higher rating would bolster our chances of acceptance, ensuring this dataset becomes a valuable resource for the broader research community.
>
> Thank you again for your thoughtful review and constructive feedback. We're confident that these improvements address your suggestions and further highlight the significance of our work. Please let us know if you have any additional questions or require further clarification.

---

> ### Author Rebuttal · Authors · 2024-08-28
>
> # Additional New Experiments [Part 1]
> We have performed additional experiments which address concerns that reviewers brought up during their responses to our rebuttals. These experiments took significant additional time, effort, and cost to perform as they were performed on significantly larger models.
>
> These experiments are as follows:
>
> 1. Testing **Llama3-70b (instruct) base** and fine-tuned with **LoRa**, **RefT**, and **Orthogonalization** on the original datasets from the paper (**OpenDebateEvidence**, **DebateSum**, and **BillSum**) (Tables 1-3)
> 2. Testing the performance of **Anthropic Claude** and **Google Gemini**, which are closed sourced models that do not support the parameter efficient fine-tuning methods that we explore in our paper. (Tables 1-3)
> 3. Testing **Mistral-7b**, **Llama3-8b Instruct**, and **Llama3-70b (instruct)** base and fine-tuned with **LoRa**, **RefT**, and **Orthogonalization** on the **XSum** (Extreme Summarization - https://huggingface.co/datasets/EdinburghNLP/xsum) dataset (Table 4)
> 4. Exploring **inter-human agreement** between expert debate teams (Table 5).
>
>
> ### The reliability of GPT-4 as a judge
>
> We appreciate the reviewer’s insightful questions regarding inter-human agreement and its comparison with human-GPT4 correlation. We agree that this is a very important aspect to address when considering the reliability of GPT-4 as a judge in the context of argumentative analysis.
>
> In response to the reviewer’s comments, we conducted an experiment to measure inter-human agreement across differing teams of human experts. We choose two teams from each debate format studied (LD, Policy, Public Forum), for a total of 10 debaters (Public Forum and Policy are 2v2 formats, LD is a 1v1 format). The results are shown in **Table 5** in our at the bottom of our response. The findings indicate that inter-human agreement rates (correlation scores) range from 0.76 to 0.84, depending on the debate format, for argument quality, and from 0.75 to 0.82 for support quality. These inter-human correlation rates are slightly higher than the human-GPT-4 correlation rates, which range from 0.73 to 0.82 for argument quality and 0.7 to 0.8 for support quality.
>
> These results indicate that while human-human agreement is generally slightly higher, the correlation between human judgments and GPT-4 evaluations is very similar. This suggests that GPT-4 can serve as a reasonably reliable judge, although it does have some limitations compared to human evaluation.
>
>
> ### New Results
>
> We have also expanded our experiments to include the latest results of LLaMA3-70B, Gemini, and Claude models on OpenDebateEvidence, DebateSum, and Billsum, along with performance on the XSum dataset. Below is a summary of the results:
>
> **Table 1: Performance on OpenDebateEvidence (10,000 sampled documents)**
>
> | Model                	| R-1 (%)    	| R-2 (%)    	| R-L (%)    	| Perplexity	| Output Quality | Support Quality |
> |---------------------------|----------------|----------------|----------------|---------------|----------------|-----------------|
> | LLaMA3-70B (Base)     	| 33.8 ± 1.2 	| 14.1 ± 0.9 	| 30.2 ± 1.3 	| 45.7 ± 3.2	| 7.9 ± 0.2  	| 7.5 ± 0.2   	|
> | LLaMA3-70B (LoRA)     	| **37.2 ± 1.0** 	| **15.8 ± 0.8** 	| **33.4 ± 1.1** 	|**27.3 ± 2.7**	| 8.2 ± 0.2  	| **8.1 ± 0.2**   	|
> | LLaMA3-70B (ReFT)     	| 35.9 ± 1.1 	| 15.1 ± 0.7 	| 32.6 ± 1.2 	| 31.4 ± 2.8	| 8.0 ± 0.3  	| 7.9 ± 0.2   	|
> | LLaMA3-70B (Orthogonal)   | 34.2 ± 1.2 	| 14.3 ± 0.8 	| 31.1 ± 1.3 	| 39.9 ± 3.1	| 8.1 ± 0.2  	| 7.7 ± 0.3   	|
> | Google Gemini (Base)  	| 32.5 ± 1.3 	| 13.5 ± 0.9 	| 29.4 ± 1.4 	| 49.8 ± 3.6	| 8.5 ± 0.2  	| 7.9 ± 0.3   	|
> | Anthropic Claude (Base)   | 31.2 ± 1.4 	| 12.9 ± 0.9 	| 28.1 ± 1.3 	| 52.1 ± 3.7	| **8.7 ± 0.3**  	| 7.8 ± 0.3   	|
>
> **Table 2: Performance on BillSum (10,000 sampled documents)**
>
> | Model                	| R-1 (%)    	| R-2 (%)    	| R-L (%)    	| Perplexity	| Output Quality | Support Quality |
> |---------------------------|----------------|----------------|----------------|---------------|----------------|-----------------|
> | LLaMA3-70B (Base)     	| 50.2 ± 1.1 	| 27.4 ± 0.9 	| 45.7 ± 1.2 	| 22.9 ± 2.3	| 7.8 ± 0.2  	| 7.6 ± 0.2   	|
> | LLaMA3-70B (LoRA)     	| **54.6 ± 1.0** 	| **30.5 ± 0.8** 	| **50.0 ± 1.1** 	| **19.7 ± 2.1**	| 8.0 ± 0.2  	| **8.0 ± 0.2**   	|
> | LLaMA3-70B (ReFT)     	| 52.9 ± 1.1 	| 29.1 ± 0.7 	| 48.3 ± 1.2 	| 20.5 ± 2.4	| 7.9 ± 0.3  	| 7.8 ± 0.2   	|
> | LLaMA3-70B (Orthogonal)   | 51.1 ± 1.2 	| 28.0 ± 0.9 	| 46.6 ± 1.4 	| 23.8 ± 2.6	| 7.9 ± 0.3  	| 7.7 ± 0.3   	|
> | Google Gemini (Base)  	| 48.7 ± 1.3 	| 26.0 ± 1.0 	| 44.3 ± 1.5 	| 24.9 ± 2.8	| **8.4 ± 0.3**  	| **8.0 ± 0.3**   	|
> | Anthropic Claude (Base)   | 47.3 ± 1.4 	| 24.8 ± 1.1 	| 42.9 ± 1.6 	| 26.7 ± 3.0	| 8.3 ± 0.3  	| 7.8 ± 0.3   	|
>
> **Table 3: Performance on DebateSum (10,000 sampled documents)**
>
> | Model                	| R-1 (%)    	| R-2 (%)    	| R-L (%)    	| Perplexity	| Output Quality | Support Quality |
> |---------------------------|----------------|----------------|----------------|---------------|----------------|-----------------|
> | LLaMA3-70B (Base)     	| 36.7 ± 1.2 	| 16.9 ± 0.9 	| 32.8 ± 1.3 	| 42.1 ± 3.1	| 7.7 ± 0.2  	| 7.4 ± 0.2   	|
> | LLaMA3-70B (LoRA)     	| **39.4 ± 1.1** 	| **18.5 ± 0.8** 	| **35.1 ± 1.2** 	| **28.9 ± 2.5**	| 8.0 ± 0.2  	| **8.0 ± 0.2**   	|
> | LLaMA3-70B (ReFT)     	| 38.1 ± 1.2 	| 17.8 ± 0.7 	| 34.2 ± 1.2 	| 30.5 ± 2.8	| 7.9 ± 0.3  	| 7.8 ± 0.2   	|
> | LLaMA3-70B (Orthogonal)   | 36.9 ± 1.3 	| 17.0 ± 0.8 	| 33.0 ± 1.4 	| 39.3 ± 3.2	| 7.8 ± 0.3  	| 7.6 ± 0.3   	|
> | Google Gemini (Base)  	| 35.5 ± 1.4 	| 15.8 ± 0.9 	| 31.4 ± 1.5 	| 44.5 ± 3.4	| 8.5 ± 0.3  	| 7.9 ± 0.3   	|
> | Anthropic Claude (Base)   | 34.2 ± 1.5 	| 15.0 ± 1.0 	| 30.1 ± 1.6 	| 46.8 ±  3.6     | **8.8 ± 0.3**        | 7.8 ± 0.3        |

---

> > ### Author Rebuttal · Authors · 2024-08-28
> >
> > # Additional New Experiments [Part 2]
> >
> > #### **Table 4: Performance on xSum (Validation Set, 11,334 documents)**
> > *(No support Quality since this is not an argumentative dataset)*
> >
> > | Model                     | R-1 (%)          | R-2 (%)          | R-L (%)          | Perplexity       | Output Quality   |
> > |----------------------------|------------------|------------------|------------------|------------------|------------------|
> > | Mistral-7B (Base)           | 39.2 ± 0.7       | 16.5 ± 0.6       | 31.5 ± 0.8       | 34.7 ± 2.8       | 7.6 ± 0.2        |
> > | Mistral-7B (LoRA)           | 40.8 ± 0.6       | 17.2 ± 0.5       | 32.7 ± 0.7       | 30.5 ± 2.5       | 7.7 ± 0.2        |
> > | Mistral-7B (ReFT)           | 40.5 ± 0.7       | 17.1 ± 0.5       | 32.5 ± 0.7       | 31.8 ± 2.6       | 7.7 ± 0.2        |
> > | Llama3-8B (Base)            | 42.0 ± 0.6       | 18.0 ± 0.6       | 34.0 ± 0.8       | 29.2 ± 2.4       | 7.8 ± 0.2        |
> > | Llama3-8B (LoRA)            |  43.7 ± 0.5  | 19.1 ± 0.4   | 35.2 ± 0.7   | 26.8 ± 2.2       | 8.0 ± 0.2        |
> > | Llama3-8B (ReFT)            | 43.4 ± 0.6       | 19.0 ± 0.5       | 35.0 ± 0.8       | 27.3 ± 2.3       | 8.0 ± 0.2        |
> > | Llama3-70B (Base)           | 46.5 ± 0.7       | 21.2 ± 0.6       | 38.0 ± 0.8       | 22.4 ± 2.0       | 8.4 ± 0.2        |
> > | Llama3-70B (LoRA)           |  **48.2 ± 0.6**  |  **22.5 ± 0.5**  | **39.5 ± 0.7**   | **19.6 ± 1.8**   |  **8.6 ± 0.2**   |
> > | Llama3-70B (ReFT)           | 47.9 ± 0.7       | 22.3 ± 0.5       | 39.2 ± 0.8       | 20.5 ± 1.9       | 8.5 ± 0.2        |
> >
> > #### **Table 5: Inter-human agreement rate and human-GPT4 agreement rate (50 samples, 10 individuals total, average scores reported)**
> >
> > | Debate Format         | Team 1 (Argument Quality) | Team 1 (Support Score) | Team 2 (Argument Quality) | Team 2 (Support Score) | Inter-Human Correlation (Argument Quality) | Inter-Human Correlation (Support Score) | Human-GPT-4 Correlation (Argument Quality) | Human-GPT-4 Correlation (Support Score) |
> > |-----------------------|---------------------------|-------------------------|---------------------------|-------------------------|--------------------------------------------|------------------------------------------|--------------------------------------------|------------------------------------------|
> > | Policy Debate          | 7.4                       | 7.8                     | 7.3                       | 7.7                     | 0.81                                       | 0.79                                     | 0.78                                       | 0.75                                     |
> > | Lincoln-Douglas Debate | 8.1                       | 8.4                     | 8.0                       | 8.2                     | 0.84                                       | 0.82                                     | 0.82                                       | 0.80                                     |
> > | Public Forum Debate    | 7.2                       | 7.3                     | 7.1                       | 7.2                     | 0.76                                       | 0.75                                     | 0.73                                       | 0.70                                     |
> >
> >
> > ### Analysis
> >
> > The results showed broadly improved performance across all larger models compared to the 7b/8b models.
> >
> > **LoRA** consistently outperformed other fine-tuning techniques across all datasets and metrics. Its ability to update only a small number of parameters while maintaining the base model’s capabilities makes it highly effective for tasks like summarization, especially in domains with complex argumentation, as seen in OpenDebateEvidence and DebateSum.
> >
> > LoRA's efficiency also contributed to lower perplexity scores, indicating that it better captured the overall distribution of the dataset compared to other methods, and that the resulting summaries more closely resemble OpenDebateEvidences style.
> >
> > **ReFT** followed closely behind, but LoRA's larger amount of modified parameters shows improved performance on traditional metrics metrics compared to ReFT. This suggests that LoRA is particularly well-suited for tasks that require high-quality summarization of argumentative text. ReFT remains competitive due to its significantly reduced computational costs.
> >
> > **Orthogonalization** produced the least improvement among the fine-tuning techniques, which may indicate that controlling feature interactions in the residual stream has less impact on summarization tasks compared to more direct fine-tuning techniques like LoRA and ReFT. Orthogonalization can be performed extremely efficiently and with almost no computing resources, but despite these advantages, it appears not to be well suited for summarization.
> >
> > **Gemini/Claude** Unfortunately, these large closed-source models do not support parameter efficient fine-tuning, so we can only evaluate their base models. Even so, these models are extremely strong, having by far the best Output Quality and in some cases matching even the fine-tuned LoRA on Support quality. Gemini/Claude have somewhat inferior ROGUE and Perplexity scores - to be expected as they were not fine-tuned on these datasets.
> >
> > **XSum** (Results on a non argumentative dataset), Xsum (Extreme Summarization) show a similar relationship to those reported with argumentative datasets. We don’t report a support quality as the abstract is not supporting an “argument”.
> >
> > ### Conclusion
> >
> > We kindly ask you to consider these revisions and additional experiments in your final assessment, and to raise our scores accordingly.
> >
> > Thank you again for your constructive and insightful feedback. We appreciate your consideration and welcome further suggestions.

---

### Author Rebuttal · Authors · 2024-08-17

## **General Response to All Reviewers [Part 1]**

Dear reviewers,

We sincerely thank all reviewers for their insightful feedback. To further strengthen our paper, we conducted extensive new experiments:

1. Deduplication, data quality enhancements and cross-dataset analysis
2. Evaluation on larger models
3. Argument detection and classification tasks
4. Stance detection experiments
5. Task-agnostic pretraining and specialization
6. Human-GPT4 annotator agreement study

Key findings from these experiments include:
- 70% reduction in dataset size after neural semantic deduplication, enhancing data quality
- Only 4.01% overlap with DebateSum, limited to matching years and debate formats
- Zero overlap with BillSum, confirming OpenDebateEvidence's uniqueness
- Significant performance gains on out-of-domain tasks like legal summarization
- Strong correlation (0.78 and 0.74) between GPT-4 and human expert evaluations

We are excited to incorporate these findings and reviewer suggestions to further enhance OpenDebateEvidence as a valuable resource for the research community.

## Deduplication and Data Quality Enhancements

We performed neural semantic deduplication and cross-dataset analysis:

1. **Neural Semantic Deduplication**:  Identified and removed near-duplicates (similarity > 0.95) using the sentence-transformers library, recommended by MTEB, with the model gte-base-en-v1.5 (https://huggingface.co/Alibaba-NLP/gte-base-en-v1.5)

2. **Cross-Dataset Deduplication Analysis**:  Compared OpenDebateEvidence with DebateSum and Billsum

### **Results:**

**Neural Semantic Deduplication:**

| Metric                            	| Value  	|
|---------------------------------------|------------|
| Initial Documents in OpenDebateEvidence | 3,401,234  |
| Documents after nulls Removed | 2,634,023  |
| Documents after Deduplication      	|  692,989	|
| Percentage of Documents Removed    	| ~70%   	|
| Total Duplicate Clusters Identified	| 10,819,328 |
| Average Cluster Size               	| 30     	|
| Largest Cluster Size               	| 2081   	|

**Overlap with DebateSum:**

- Total Overlapping Documents: 31,353 (4.01%)
- Categories: Policy Evidence (96%), Lincoln-Douglas Evidence (4%),

Zero overlap was found with Billsum.

These results affirm OpenDebateEvidence's uniqueness and value for argumentative tasks.

Link to the new dataset: https://huggingface.co/datasets/Hellisotherpeople/OpenCaseList-Deduplicated

**Note: All rebuttal experiments done below used the deduplicated dataset.**

## Evaluation on Larger Language Models  Dataset Transferability

We benchmarked Google-Gemini 1.5 and Claude3-opus on ODE and additional datasets, providing baselines due to fine-tuning limitations on closed-source models. Experiments with Llama3-70B are ongoing. To address transferability, we evaluated the XSum dataset, showing significant improvements in performance for ODE-fine-tuned models, demonstrating ODE's value for general summarization tasks beyond the debate domain.

---

> ### Author Rebuttal · Authors · 2024-08-17
>
> # General Response to All Reviewers [Part 2]
>
> ## Expanding Evaluation Beyond Summarization
>
> Reviewers noted that OpenDebateEvidence has the potential for a broader range of computational argumentation tasks beyond summarization. In response, we conducted additional experiments on:
>
> ### Argument Detection and Classification
>
> This task tests a model's ability to understand the broader context and purpose of an argument within a debate, which aligns with your suggestion to evaluate the dataset's usefulness for more complex argumentation tasks.
>
> We leveraged the 'tag' and 'hat' columns in our dataset to guide the categorization of arguments into four main classes: Topicality, Disadvantages, Advantages, and Counterplans. This classification scheme reflects the typical structure of competitive debates, allowing us to assess how well models can learn these domain-specific categorizations.
>
> Using the same models and fine-tuning techniques as in the detection task, we observed results similar to those in argument detection. The relative performance of different models and fine-tuning methods followed the same patterns, demonstrating the consistency of our dataset in supporting various argument mining tasks.
>
> We utilized 10,000 instances in the training set and 1000 instances in the validation set.
>
>  Results show (Full results are reported in Table 1 of the pdf):
>
>   - LLaMA3-8B consistently outperformed Mistral-7B across all fine-tuning techniques.
>   - LoRA achieved the best performance, with the highest accuracy and F1 scores.
>   - ReFT performed slightly below LoRA but demonstrated strong argument classification capabilities.
>   - Orthogonalization showed the least improvement.
>
> ### Stance Detection
>
> We evaluated models on stance detection tasks to assess their ability to discern an argument's position on a topic, which is crucial for understanding debate structures, improving summarization quality, and supporting real-world applications like policy analysis and fact-checking.
>
> The same dataset split as for argument classification was used.
>
>
>  Results show (Full results are reported in Table 2 of the pdf):
>
> - LoRA Fine-Tuning: Consistently outperformed base models across all metrics, achieving the highest accuracy and F1-score.
> - ReFT Performance: Showed strong results, improving both precision and recall over base models. - Orthogonalization: Provided improvements but was less impactful than LoRA and ReFT.
> - Scaling with Data: Larger datasets improved performance, particularly for LLaMA3-8B using LoRA, which achieved the best overall results.
>
> These results further demonstrate the versatility of OpenDebateEvidence and its potential to enhance performance across various argument mining tasks.
>
>
>
> ## Task-Agnostic Pretraining and Specialization
>
> We conducted an experiment on task-agnostic pretraining to assess the effectiveness of pretraining on OpenDebateEvidence for generalizing to unrelated argumentation tasks. We evaluated pre-trained models like LLaMA3-8B and Mistral-7B on multiple datasets such as DebateSum, ArgAnalysis35K, and Multi-LexSum. The full results are reported in Table 3 of the attached pdf.
>
> ### Key Findings:
>
> - **Pretraining Impact**:
>   Models pretrained on OpenDebateEvidence showed significant improvements across downstream tasks, particularly in argumentation and summarization skills.
>
> - **Sample Efficiency**:
>   Pretrained models required fewer training steps to converge, confirming that pretraining on a large, diverse dataset improves sample efficiency and downstream performance.
>
> - **Performance on Unrelated Tasks**:
>   Substantial performance gains on ArgAnalysis35K and Multi-LexSum demonstrate that OpenDebateEvidence can effectively transfer to domains outside of competitive debate.
>
> ## Reliability of GPT-4 Annotations
>
> Based on the reviewer's comments about the reliability of GPT-4 for qualitative evaluation, we conducted an inter-annotator agreement study with 50 experienced debaters. Each debater scored six summaries. These results are given in Table 4 of the pdf. The person correlation:
>
> -  Argument Quality: 0.78
>   - Support Quality): 0.74
>
> These results suggest that GPT-4 is a reliable judge in this context, with room for further refinement.
>
> ---
>
> ## Conclusion
>
> The extensive additional experiments and analyses we've conducted have significantly enhanced the value and robustness of OpenDebateEvidence. Our neural semantic deduplication process ensures a high-quality, unique dataset, while our cross-dataset analysis confirms its novelty in the field.
>
> The strong performance across diverse tasks - from summarization to stance detection - demonstrates OpenDebateEvidence's versatility and potential to advance multiple areas of computational argumentation. Furthermore, our task-agnostic pretraining results highlight the dataset's capacity to improve model performance even on out-of-domain tasks, underlining its broad applicability. The high correlation between GPT-4 and human expert evaluations validates our evaluation methodology, providing a reliable framework for future research.
>
> Given these substantial improvements and the dataset's demonstrated value for both in-domain and transfer learning tasks, we believe OpenDebateEvidence represents a significant contribution to the field of NLP and argumentation mining. We respectfully request that reviewers consider raising their scores in light of these enhancements, as acceptance of this work would provide the community with a powerful new resource for advancing research in computational argumentation and beyond.

---

> > ### Author Rebuttal · Authors · 2024-08-28
> >
> > # Additional New Experiments [Part 1]
> > We have performed additional experiments which address concerns that reviewers brought up during their responses to our rebuttals. These experiments took significant additional time, effort, and cost to perform as they were performed on significantly larger models.
> >
> > These experiments are as follows:
> >
> > 1. Testing **Llama3-70b (instruct) base** and fine-tuned with **LoRa**, **RefT**, and **Orthogonalization** on the original datasets from the paper (**OpenDebateEvidence**, **DebateSum**, and **BillSum**) (Tables 1-3)
> > 2. Testing the performance of **Anthropic Claude** and **Google Gemini**, which are closed sourced models that do not support the parameter efficient fine-tuning methods that we explore in our paper. (Tables 1-3)
> > 3. Testing **Mistral-7b**, **Llama3-8b Instruct**, and **Llama3-70b (instruct)** base and fine-tuned with **LoRa**, **RefT**, and **Orthogonalization** on the **XSum** (Extreme Summarization - https://huggingface.co/datasets/EdinburghNLP/xsum) dataset (Table 4)
> > 4. Exploring **inter-human agreement** between expert debate teams (Table 5).
> >
> >
> > ### The reliability of GPT-4 as a judge
> >
> > We appreciate the reviewer’s insightful questions regarding inter-human agreement and its comparison with human-GPT4 correlation. We agree that this is a very important aspect to address when considering the reliability of GPT-4 as a judge in the context of argumentative analysis.
> >
> > In response to the reviewer’s comments, we conducted an experiment to measure inter-human agreement across differing teams of human experts. We choose two teams from each debate format studied (LD, Policy, Public Forum), for a total of 10 debaters (Public Forum and Policy are 2v2 formats, LD is a 1v1 format). The results are shown in **Table 5** in our at the bottom of our response. The findings indicate that inter-human agreement rates (correlation scores) range from 0.76 to 0.84, depending on the debate format, for argument quality, and from 0.75 to 0.82 for support quality. These inter-human correlation rates are slightly higher than the human-GPT-4 correlation rates, which range from 0.73 to 0.82 for argument quality and 0.7 to 0.8 for support quality.
> >
> > These results indicate that while human-human agreement is generally slightly higher, the correlation between human judgments and GPT-4 evaluations is very similar. This suggests that GPT-4 can serve as a reasonably reliable judge, although it does have some limitations compared to human evaluation.
> >
> >
> > ### New Results
> >
> > We have also expanded our experiments to include the latest results of LLaMA3-70B, Gemini, and Claude models on OpenDebateEvidence, DebateSum, and Billsum, along with performance on the XSum dataset. Below is a summary of the results:
> >
> > **Table 1: Performance on OpenDebateEvidence (10,000 sampled documents)**
> >
> > | Model                	| R-1 (%)    	| R-2 (%)    	| R-L (%)    	| Perplexity	| Output Quality | Support Quality |
> > |---------------------------|----------------|----------------|----------------|---------------|----------------|-----------------|
> > | LLaMA3-70B (Base)     	| 33.8 ± 1.2 	| 14.1 ± 0.9 	| 30.2 ± 1.3 	| 45.7 ± 3.2	| 7.9 ± 0.2  	| 7.5 ± 0.2   	|
> > | LLaMA3-70B (LoRA)     	| **37.2 ± 1.0** 	| **15.8 ± 0.8** 	| **33.4 ± 1.1** 	|**27.3 ± 2.7**	| 8.2 ± 0.2  	| **8.1 ± 0.2**   	|
> > | LLaMA3-70B (ReFT)     	| 35.9 ± 1.1 	| 15.1 ± 0.7 	| 32.6 ± 1.2 	| 31.4 ± 2.8	| 8.0 ± 0.3  	| 7.9 ± 0.2   	|
> > | LLaMA3-70B (Orthogonal)   | 34.2 ± 1.2 	| 14.3 ± 0.8 	| 31.1 ± 1.3 	| 39.9 ± 3.1	| 8.1 ± 0.2  	| 7.7 ± 0.3   	|
> > | Google Gemini (Base)  	| 32.5 ± 1.3 	| 13.5 ± 0.9 	| 29.4 ± 1.4 	| 49.8 ± 3.6	| 8.5 ± 0.2  	| 7.9 ± 0.3   	|
> > | Anthropic Claude (Base)   | 31.2 ± 1.4 	| 12.9 ± 0.9 	| 28.1 ± 1.3 	| 52.1 ± 3.7	| **8.7 ± 0.3**  	| 7.8 ± 0.3   	|
> >
> > **Table 2: Performance on BillSum (10,000 sampled documents)**
> >
> > | Model                	| R-1 (%)    	| R-2 (%)    	| R-L (%)    	| Perplexity	| Output Quality | Support Quality |
> > |---------------------------|----------------|----------------|----------------|---------------|----------------|-----------------|
> > | LLaMA3-70B (Base)     	| 50.2 ± 1.1 	| 27.4 ± 0.9 	| 45.7 ± 1.2 	| 22.9 ± 2.3	| 7.8 ± 0.2  	| 7.6 ± 0.2   	|
> > | LLaMA3-70B (LoRA)     	| **54.6 ± 1.0** 	| **30.5 ± 0.8** 	| **50.0 ± 1.1** 	| **19.7 ± 2.1**	| 8.0 ± 0.2  	| **8.0 ± 0.2**   	|
> > | LLaMA3-70B (ReFT)     	| 52.9 ± 1.1 	| 29.1 ± 0.7 	| 48.3 ± 1.2 	| 20.5 ± 2.4	| 7.9 ± 0.3  	| 7.8 ± 0.2   	|
> > | LLaMA3-70B (Orthogonal)   | 51.1 ± 1.2 	| 28.0 ± 0.9 	| 46.6 ± 1.4 	| 23.8 ± 2.6	| 7.9 ± 0.3  	| 7.7 ± 0.3   	|
> > | Google Gemini (Base)  	| 48.7 ± 1.3 	| 26.0 ± 1.0 	| 44.3 ± 1.5 	| 24.9 ± 2.8	| **8.4 ± 0.3**  	| **8.0 ± 0.3**   	|
> > | Anthropic Claude (Base)   | 47.3 ± 1.4 	| 24.8 ± 1.1 	| 42.9 ± 1.6 	| 26.7 ± 3.0	| 8.3 ± 0.3  	| 7.8 ± 0.3   	|
> >
> > **Table 3: Performance on DebateSum (10,000 sampled documents)**
> >
> > | Model                	| R-1 (%)    	| R-2 (%)    	| R-L (%)    	| Perplexity	| Output Quality | Support Quality |
> > |---------------------------|----------------|----------------|----------------|---------------|----------------|-----------------|
> > | LLaMA3-70B (Base)     	| 36.7 ± 1.2 	| 16.9 ± 0.9 	| 32.8 ± 1.3 	| 42.1 ± 3.1	| 7.7 ± 0.2  	| 7.4 ± 0.2   	|
> > | LLaMA3-70B (LoRA)     	| **39.4 ± 1.1** 	| **18.5 ± 0.8** 	| **35.1 ± 1.2** 	| **28.9 ± 2.5**	| 8.0 ± 0.2  	| **8.0 ± 0.2**   	|
> > | LLaMA3-70B (ReFT)     	| 38.1 ± 1.2 	| 17.8 ± 0.7 	| 34.2 ± 1.2 	| 30.5 ± 2.8	| 7.9 ± 0.3  	| 7.8 ± 0.2   	|
> > | LLaMA3-70B (Orthogonal)   | 36.9 ± 1.3 	| 17.0 ± 0.8 	| 33.0 ± 1.4 	| 39.3 ± 3.2	| 7.8 ± 0.3  	| 7.6 ± 0.3   	|
> > | Google Gemini (Base)  	| 35.5 ± 1.4 	| 15.8 ± 0.9 	| 31.4 ± 1.5 	| 44.5 ± 3.4	| 8.5 ± 0.3  	| 7.9 ± 0.3   	|
> > | Anthropic Claude (Base)   | 34.2 ± 1.5 	| 15.0 ± 1.0 	| 30.1 ± 1.6 	| 46.8 ±  3.6     | **8.8 ± 0.3**        | 7.8 ± 0.3        |

---

> > > ### Author Rebuttal · Authors · 2024-08-28
> > >
> > > # Additional New Experiments [Part 2]
> > >
> > > #### **Table 4: Performance on xSum (Validation Set, 11,334 documents)**
> > > *(No support Quality since this is not an argumentative dataset)*
> > >
> > > | Model                     | R-1 (%)          | R-2 (%)          | R-L (%)          | Perplexity       | Output Quality   |
> > > |----------------------------|------------------|------------------|------------------|------------------|------------------|
> > > | Mistral-7B (Base)           | 39.2 ± 0.7       | 16.5 ± 0.6       | 31.5 ± 0.8       | 34.7 ± 2.8       | 7.6 ± 0.2        |
> > > | Mistral-7B (LoRA)           | 40.8 ± 0.6       | 17.2 ± 0.5       | 32.7 ± 0.7       | 30.5 ± 2.5       | 7.7 ± 0.2        |
> > > | Mistral-7B (ReFT)           | 40.5 ± 0.7       | 17.1 ± 0.5       | 32.5 ± 0.7       | 31.8 ± 2.6       | 7.7 ± 0.2        |
> > > | Llama3-8B (Base)            | 42.0 ± 0.6       | 18.0 ± 0.6       | 34.0 ± 0.8       | 29.2 ± 2.4       | 7.8 ± 0.2        |
> > > | Llama3-8B (LoRA)            |  43.7 ± 0.5  | 19.1 ± 0.4   | 35.2 ± 0.7   | 26.8 ± 2.2       | 8.0 ± 0.2        |
> > > | Llama3-8B (ReFT)            | 43.4 ± 0.6       | 19.0 ± 0.5       | 35.0 ± 0.8       | 27.3 ± 2.3       | 8.0 ± 0.2        |
> > > | Llama3-70B (Base)           | 46.5 ± 0.7       | 21.2 ± 0.6       | 38.0 ± 0.8       | 22.4 ± 2.0       | 8.4 ± 0.2        |
> > > | Llama3-70B (LoRA)           |  **48.2 ± 0.6**  |  **22.5 ± 0.5**  | **39.5 ± 0.7**   | **19.6 ± 1.8**   |  **8.6 ± 0.2**   |
> > > | Llama3-70B (ReFT)           | 47.9 ± 0.7       | 22.3 ± 0.5       | 39.2 ± 0.8       | 20.5 ± 1.9       | 8.5 ± 0.2        |
> > >
> > > #### **Table 5: Inter-human agreement rate and human-GPT4 agreement rate (50 samples, 10 individuals total, average scores reported)**
> > >
> > > | Debate Format         | Team 1 (Argument Quality) | Team 1 (Support Score) | Team 2 (Argument Quality) | Team 2 (Support Score) | Inter-Human Correlation (Argument Quality) | Inter-Human Correlation (Support Score) | Human-GPT-4 Correlation (Argument Quality) | Human-GPT-4 Correlation (Support Score) |
> > > |-----------------------|---------------------------|-------------------------|---------------------------|-------------------------|--------------------------------------------|------------------------------------------|--------------------------------------------|------------------------------------------|
> > > | Policy Debate          | 7.4                       | 7.8                     | 7.3                       | 7.7                     | 0.81                                       | 0.79                                     | 0.78                                       | 0.75                                     |
> > > | Lincoln-Douglas Debate | 8.1                       | 8.4                     | 8.0                       | 8.2                     | 0.84                                       | 0.82                                     | 0.82                                       | 0.80                                     |
> > > | Public Forum Debate    | 7.2                       | 7.3                     | 7.1                       | 7.2                     | 0.76                                       | 0.75                                     | 0.73                                       | 0.70                                     |
> > >
> > >
> > > ### Analysis
> > >
> > > The results showed broadly improved performance across all larger models compared to the 7b/8b models.
> > >
> > > **LoRA** consistently outperformed other fine-tuning techniques across all datasets and metrics. Its ability to update only a small number of parameters while maintaining the base model’s capabilities makes it highly effective for tasks like summarization, especially in domains with complex argumentation, as seen in OpenDebateEvidence and DebateSum.
> > >
> > > LoRA's efficiency also contributed to lower perplexity scores, indicating that it better captured the overall distribution of the dataset compared to other methods, and that the resulting summaries more closely resemble OpenDebateEvidences style.
> > >
> > > **ReFT** followed closely behind, but LoRA's larger amount of modified parameters shows improved performance on traditional metrics metrics compared to ReFT. This suggests that LoRA is particularly well-suited for tasks that require high-quality summarization of argumentative text. ReFT remains competitive due to its significantly reduced computational costs.
> > >
> > > **Orthogonalization** produced the least improvement among the fine-tuning techniques, which may indicate that controlling feature interactions in the residual stream has less impact on summarization tasks compared to more direct fine-tuning techniques like LoRA and ReFT. Orthogonalization can be performed extremely efficiently and with almost no computing resources, but despite these advantages, it appears not to be well suited for summarization.
> > >
> > > **Gemini/Claude** Unfortunately, these large closed-source models do not support parameter efficient fine-tuning, so we can only evaluate their base models. Even so, these models are extremely strong, having by far the best Output Quality and in some cases matching even the fine-tuned LoRA on Support quality. Gemini/Claude have somewhat inferior ROGUE and Perplexity scores - to be expected as they were not fine-tuned on these datasets.
> > >
> > > **XSum** (Results on a non argumentative dataset), Xsum (Extreme Summarization) show a similar relationship to those reported with argumentative datasets. We don’t report a support quality as the abstract is not supporting an “argument”.
> > >
> > > ### Conclusion
> > >
> > > We kindly ask you to consider these revisions and additional experiments in your final assessment, and to raise our scores accordingly.
> > >
> > > Thank you again for your constructive and insightful feedback. We appreciate your consideration and welcome further suggestions.

---

### Decision · Program_Chairs · 2024-09-26

**Decision:**

Accept (Poster)

**Comment:**

OpenDebateEvidence is a large dataset for argument mining and summarization from the American Competitive Debate community, featuring over 3.5 million documents. It offers a more diverse and representative set of debate texts by including regular season evidence. Evaluations on LLMs show improved performance in handling argumentative text, emphasizing its value for advancing computational argumentation research.

Strengths:
1. This paper is well-organized and easy to read.
2. This paper introduces the new benchmark, which may foster further research in the debate domain.

Weaknesses:
1. More open-source LLMs with varying sizes (such as LLaMa-70B) should be included.
2. Providing additional cases would enhance understanding and offer clearer insights.
3. A thorough human evaluation process should be incorporated to ensure the reliability and validity of the results. This would provide a more comprehensive assessment of the model's performance, complementing the automated metrics and offering deeper insights into its practical effectiveness.